# On the Use of NB-IoT over GEO Satellite Systems with Time-Packed Optical Feeder Links for Over-the-Air Firmware/Software Updates of Machine-Type Terminals

**DOI:** 10.3390/s21123952

**Published:** 2021-06-08

**Authors:** Joan Bas, Alexis A. Dowhuszko

**Affiliations:** 1Centre Tecnològic de Telecomunicacions de Catalunya (CTTC/CERCA), Department of Array and Multi-Sensor Processing, Av. Karl Friedriech Gauss 7—Building B4, 08860 Castelldefels, Spain; 2Department of Communications and Networking, Aalto University, 02150 Espoo, Finland; alexis.dowhuszko@aalto.fi

**Keywords:** NB-IoT, DVB-S2(X), high-throughput satellite, optical feeder link, over-the-air updates, time-packing, decode-and-forward, scintillation, beam-wander, convolutional coding

## Abstract

The verticals of 5G, such as the automotive, smart grid, and smart cities sectors, will bring new sensors and IoT devices requiring Internet connectivity. Most of these machine-type terminals will be sparsely distributed, covering a very large geographical area and, from time to time, will have to update their software, firmware, and/or other relevant data. Given this situation, one viable solution to implement the “Over-the-Air” update of these IoT terminals can be done with the aid of GEO satellite systems. However, due to the ultra-dense radio frequency reuse factor that contemporary High-Throughput Satellite (HTS) systems implement in the access link to serve the IoT terminals, the use of a time-packed Free Space Optical (FSO) link represents a practical solution to avoid the bottleneck that the satellite gateway experiences in the feeder link. The performance of both Detect-and-Forward and Decode-and-Forward relaying strategies are studied, assuming that the single-carrier *M*-PAM symbols that are transmitted on the optical feeder link are mapped into *M*-QAM symbols that modulate the multiple sub-carriers of the OFDM-based radio access link. In addition, the benefits of encapsulating the NB-IoT frames into DVB-S2(X) satellite frames is also analyzed in detail. The effects of the impairments introduced in both the optical feeder and radio access links are characterized in detail, and the end-to-end error correction capabilities of the Modulation and Coding Schemes (MCS) defined in the contemporary releases of the NB-IoT and DVB-S2(X) standards are studied for different working regimes.

## 1. Introduction

In the forthcoming years, an increased data rate capacity will be needed to provide enhanced Mobile Broadband (eMBB) and massive Machine-Type Communications (mMTC) fueled, among other reasons, by the large demand of video transmissions and IoT communications that are foreseen in future vertical services [1]. Specifically, according to CISCO, it is expected that Machine-to-Machine (M2M) connections will ammount to 14.7 billion by 2023. Applications such as home automation, home security, video surveillance, connected white goods, and tracking are expected to represent almost half of the total M2M connections by 2023 [2]. Furthermore, the M2M services that are currently experiencing the fastest growth are connected car applications, such as fleet management, in-vehicle entertainment, emergency calling, vehicle diagnostics, and navigation, with a Compound Annual Growing Rate (CAGR) in the order of 30%. All these applications share the same requirement, known as Over-The-Air (OTA) programming, to update the software (e.g., maps for navigation), security keys (e.g., for cryptography in the IoT devices), among other firmware updates.

Lately, 3GPP has started its studies to analyze the feasibility of integrating satellite networks into its mobile communications systems [3]. Therefore, until these studies are completed, OTA services provided over satellites are based on proprietary solutions owned by the satellite operators. Regarding the feeder links, the dominant State-of-Art (SoA) technology for implementing them nowadays is based on multi-beam reflect arrays in the Ka/Ku radio frequency band [4,5], using the DVB-S2 standard. However, there are many satellites in different orbits allocated in the Ka/Ku bands, and limited bandwidth is currently available for new deployments, particularly for 5G networks and beyond [6]. Note that 6G systems will have to support terminals with a mobility speed of up to 1000 km/h, a connectivity density of $10^{7}$ devices/km^2^, and peak data dates in downlink of up to 1 Tera-bit-per-second (Tbps) to name a few of its requirements [7,8,9]. This means that the capacity of a satellite feeder link must be increased, in order to avoid possible bottlenecks. Non-terrestrial networks, particularly satellite ones, will have to play a key role in the next generation of mobile communication systems, integrating airborne, terrestrial, and satellite networks to support in-flight connectivity. Then, multiple technologies, not only radio-based ones, must be used to face the constrains that are foreseen for 6G.

In practical terms, the growth of OTA applications that is expected in the near future implies that the downlink channel (i.e., from the base station to the IoT terminals) will have to be improved to support the forecast data traffic demand. Furthermore, given that machine-type terminals may be sparsely deployed in large geographical areas, then the use of satellite network would be an excellent option to enable reliable IoT connectivity in a global scale. Towards this regard, the advent of Very High Throughput Satellite (VHTS) systems will allow one to achieve a total network capacity of few Tbps [10,11]. In line with this, satellite operators are starting to study the viability of deploying optical feeder links for Beyond 5G applications in GEO [6], where primary research analyses have been already conducted [12]. Recent experimental studies in single optical LEO-to-ground links have also been reported in [13]. Larger communication bandwidths and free access to spectrum are the key points in favor of optical wireless communications. Unfortunately, both radio and optical wireless communications suffer from channel impairments. Therefore, from a long-term perspective, hybrid solutions combining both technologies should be implemented in the forthcoming generations of satellite networks [14] to increase the capacity of the feeder link.

Regarding the potential techniques to increase the spectral efficiency of the optical feeder link, e.g., beamforming [15], NOMA [16], and frequency reuse [17], this paper resorts to the ones based on shrinking the transmitted pulses. The so-called Faster-Than-Nyquist or time-packed techniques. Initially proposed by Mazo in the 70s [18], has been proposed as a a potential technique for increasing the spectral efficiency of Beyond 5G systems [19]. From the satellite point of view, the possibility of reducing the transmission time without augmenting the transmission bandwidth fits very well for overcoming the drawbacks of the optical channel, which heavily depend on atmospheric conditions such as clouds [6]. By doing so, the link layer may adjust the transmission time of the frames without reducing the number of symbols to transmit according to weather forecasts [20].

This paper focuses on using optical feeder links for GEO satellite networks. Since initial studies about the introduction of satellites into the 3GPP landscape have focused on low-latency applications, most of these satellite communication scenarios are dominated by the use of Low Earth Orbit (LEO) satellite constellations [21]. However, due to the longer space-to-ground link distances of GEO satellites, more spectral-efficient techniques should be introduced in their architectures. Furthermore, observing the evolution that 3GPP standards have experienced in the past, first integrating vehicular and railway networks into 5G [8] and then airbone and satellites into 6G [7], it is highly probable that at some point in the future, extra-terrestrial communications will also be integrated into the 3GPP ecosystem. In this latter scenario, GEO satellites are in a good position for relaying data from the Earth to the outer space, as discussed in the Moonlight initiative from ESA [22]. Therefore, this paper studies the approach that GEO satellites should use to forward the information that they receive from the satellite gateway [23].

In the “ideal” case of a GEO satellite that implements a fully regenerative payload, the optical feeder link would be terminated in the satellite, and a robust modulation and coding scheme should be selected to address the bit error bursts that the turbulent optical satellite wireless channel introduce [24]. In the transparent non-regenerative “bent-pipe” solutions, on the other hand, the instantaneous value of the radio signal is used to modulate the intensity of the optical carrier of the feeder link Laser Diode (LD) with the aid of an external Match-Zehnder Modulator (MZM) [25]. Moreover, if time-packing encoding is applied on the real-valued electrical signal that is used to modulate the intensity of the LD beam, the data throughput of the optical feeder link can be increased even further, without the necessity of using a wider communication bandwidth. This effect is obtained by shrinking the separation between adjacent transmitted pulses [26], mitigating part of the Inter-Symbol Interference (ISI) power that time-packing introduces with the aid of a linear equalizer that is placed on-board the GEO satellite before symbol detection [27]. It is important to highlight that the impact of the residual ISI, which remains in the forward link after the GEO satellite relaying, can be further mitigated with the proper selection of the Modulation and Coding Schemes (MCS) to communicate with the NB-IoT terminal [28].

The forward link of a satellite system can be divided into two parts, namely: (i) The optical feeder link (uplink), from the ground station to the GEO satellite, and (ii) the radio access link (downlink), from the GEO satellite to the NB-IoT user terminals. Therefore, and in order to improve the achievable throughput of the forward direction of communication of the GEO satellite system under different working conditions, this paper studies three different relaying architectures, namely: (1) *Detect-and-Forward* (a non-regenerative strategy), where the GEO satellite only detects the symbols of the NB-IoT frames that modulate the intensity of the optical beam, and forwards them to the NB-IoT devices after *M*-PAM to *M*-QAM mapping; (2) *Decode-and-Forward with NB-IoT* (a regenerative strategy), where the GEO satellite detects and decodes the symbols of the NB-IoT frames that are transported on the optical feeder link, and forwards them to the IoT devices after the NB-IoT frame regeneration for the downlink radio transmission is over; (3) *Decode-and-Forward with NB-IoT/DVB-S2(X)* (a regenerative strategy), where the NB-IoT frames are encapsulated into DVB-S2(X) satellite frames for uplink transmission and, in the GEO satellite, the DVB-S2(X) decoding is performed to recover the NB-IoT frame that is then transmitted to the NB-IoT terminals. We note that the previously listed relaying architectures are also applicable to LEO and Medium Earth Orbit (MEO) satellites. However, to adapt the study presented in this paper from GEO satellites with fixed positions in the sky to LEO/MEO satellites that change their position on the sky at different speeds, the modeling of the optical channel and its link budget must be adapted accordingly, reducing the (distance-dependent) path loss attenuation that is experienced at lower orbits but adding an additional effect of the atmosphere at low-elevation angles, as well as pointing errors that may be incurred when the satellite is moving.

The remaining part of this article has been structured as follows: Section 2 summarizes the key concepts to model the MCS defined in the NB-IoT standard, the three proposed relaying architectures using GEO satellites with optical feeder links (one non-regenerative and two regenerative), and the details of the time-packing equalization and low-complexity Log Likelihood Ratio (LLR) computation for soft decoding. Section 3 studies the effect of the turbulent atmosphere in the optical feeder link, with emphasis on the beam wander and scintillation that is introduced in the uplink transmission. Section 4 presents the simulation set up and the figures in terms of end-to-end Block Error Rate (BLER) and throughput. Finally, conclusions are drawn in Section 5.

## 2. System Model

This section summarizes the key technological concepts that are needed to model the link level of the NB-IoT communication, as well as the different relaying architectures that could be used on-board the GEO satellite to interface the optical wireless signal (feeder link) into the radio wireless signal (access link) that is forwarded by the GEO satellite to the NB-IoT terminals.

### 2.1. NB-IoT Signal Format for the Satellite Forward Link

NB-IoT has been developed by 3GPP to cope with the large demand on IoT connectivity that is foreseen by the designers of the future generations of mobile communication standards (i.e., 5G and beyond). NB-IoT has been conceived to be deployed in three different configurations or typologies, which are compatible with the spectrum allocation (channelization) that is used in contemporary mobile communication standards such as GSM (2G) and LTE (4G). An overview of these deployment typologies, which are known as stand-alone, in-band, and guard-band, can be appreciated in Figure 1.

In the stand-alone deployment, the NB-IoT signal occupies the bandwidth that corresponds to one (or few) of the 200-kHz GSM radio channels of the 2G radio spectrum; note that this strategy is suitable for the re-farming process of the GSM bands. In the in-band deployment, the NB-IoT signal is placed on the radio spectrum that corresponds to one (or few) Physical Resource Blocks (PRB) of an LTE carrier, where each PRB spans over 180 kHz of bandwidth and is formed by 12 sub-carriers of a 15-kHz bandwidth each. Finally, in the third typology of deployment known as guard-band deployment, the NB-IoT signal is placed on the guard bands that are reserved to prevent adjacent-channel interference between LTE carriers. Note that these strategies of deployment do not imply any additional cost and time to enter in service, provided the operator owns a licence either in the 2G/4G radio bands.

According to 3GPP standardization, both NB-IoT uplink and downlink transmissions occupy a communication bandwidth of 180 kHz in the radio portion of the electromagnetic spectrum. Moreover, as the downlink of NB-IoT defines the technology to communicate from the base station (eNB) to the user terminal (IoT device), we focus on this direction of communication to design the radio frame that should be used in the forward link of the IoT satellite system. Specifically, the downlink of NB-IoT uses few 15 kHz sub-carriers, providing a sampling time unit of $T_{s}=1/\left(15000\times 2048\right)$ s., which is identical to the one used in the LTE standard. Similarly, the time slot duration in NB-IoT is $T_{\mathrm{slot}}=15360\times T_{s}\;=\;0.5$ ms [29]. Two consecutive NB-IoT time slots constitute a subframe, which spans 1 ms. Similarly to LTE, a group of 10 subframes with total duration $T_{\mathrm{frame}}=10\times 2\times T_{s}=10$ ms constitutes a NB-IoT frame.

The NB-IoT standard enables to repeat the transmission of the same information (block data) up to 2048 times, in order to extend the coverage range and increase the reliability of the data communication [29]. However, the higher the number of repetitions that are performed, the lower the spectral efficiency of the data communication that takes place. The NB-IoT link selects the Transport Block Size (TBS) on a MAC layer from a variety of sizes, which range from 2 bytes (16 bits) up to 317 bytes (2536 bits) [30]. The number of Modulation and Coding Schemes (MCS) that NB-IoT supports is equal to 14, and the combination of a number of subframes and MCS to be used for communication determines the code rate of the NB-IoT transmission. Regarding the error control coding, the downlink of NB-IoT uses a 1/3 tail-biting convolutional encoding mother code [31]. This encoding procedure is formed by three generator polynomials, which are known as the $G_{0}=133$, $G_{1}=177$, and $G_{2}=165$ polynomials in the octal notation (see Figure 2 for more details). Then, after channel encoding, data rate matching is utilized to obtain the desired code rate. This rate-matching procedure is a puncturing process to obtain code rates that are higher than the one provided by the mother code (i.e., code rates higher than 1/3). However, in order to obtain code rates that are lower than the one provided by the mother code, the NB-IoT matching procedure combines block data repetition with puncturing [29].

The downlink of NB-IoT is formed by four channels, namely: Narrowband Physical Downlink Control Channel (NPDCCH), Narrowband Physical Downlink Shared Channel (NPDSCH), Narrowband Physical Broadcast Channel (NPBCH), and Narrowband Synchronization Signals (NPSS/NSSS) [32]. The first channel, the NPDCCH, is used for the control plane and provides the scheduling information for the downlink and uplink data channels. The second channel, the NPDSCH, is used for the data plane and for paging, and contains dedicated and common downlink data. The third channel, the NPBCH, contains information for the initial acquisition conveying information about the cell parameters. The NPDCCH, NPDSCH, and NPBCH channels are QPSK modulated [31]. Finally, the NPSS/NSSS signals are used to perform the cell search, time and frequency synchronization, and cell identity detection procedures, which are modulated using Zadoff–Chu sequences. In this paper, however, we focus the attention on the study of the QPSK-modulated downlink channels.

Finally, the forward link of the satellite relaying system is formed by two links, namely: (i) The link from the gateway to the satellite, the so-called feeder link and (ii) the link from the satellite to the corresponding IoT terminal, the so-called access link. The satellite gateway aggregates the NB-IoT downlink channels and send them to the corresponding satellite beam of the access link. At the satellite, the received data is switched to the access beam of the target IoT device. As expected, the satellite architecture enables to increase the throughput of the system. The following section gives further details on the satellite architectural options that are here studied to forward data from the gateway to the NB-IoT devices.

### 2.2. Architectures for Forwarding NB-IoT Frames over a Satellite Relaying Node

In the coming years, the feeder links of satellite systems will start to introduce optical wireless technology to cope with the capacity demand that new 5G/6G services will require, such a the OTA applications [6]. However, the throughput of the optical links can be increased even further by adding more advanced spectral-efficient communication techniques. Specifically, this paper considers that time-packing can be one the enablers to enlarge the end-to-end throughput of HTS systems. Nevertheless, the architecture of the optical receiver placed on the satellite payload is also important to increase the HTS throughput. Some of these architectures have been already considered in the deployment of non-terrestrial networks, resulting in bent-pipe and fully-regenerative satellite configurations [3]. However, this paper extends the possibilities of fully-regenerative satellites (i.e., Decode-and-Forward) by considering the partial regeneration of NB-IoT frames until the modulation level (i.e., Detect-and-Forward), and by considering the encapsulation of the NB-IoT modulated symbols in DVB-S2(X) frames. Specifically, this paper analyzes the throughput of the following satellite architectures, namely: (i) Case 1: NB-IoT modulated symbols detected at the satellite and forwarded to the corresponding IoT terminal; (ii) Case 2: NB-IoT frames decoded until the bit-level, re-encoded, re-mapped at the satellite, and re-sent to the IoT terminal; and (iii) Case 3: NB-IoT modulated symbols are encapsulated in DVB-S2(X) frames and re-sent to the IoT terminal (see Figure 3 for more details).

#### 2.2.1. Case 1: Detect-and-Forward Relaying of NB-IoT Frames

This strategy is similar to the one used in transparent (bent-pipe) satellite architectures, but with the difference that the NB-IoT amplitude-modulated symbols in the optical feeder link are time-packed. The Inter-Symbol Interference (ISI) that time-packing introduces is removed in the satellite (see Section 2.3) and, after that, the 4-PAM modulated symbols of the NB-IoT signal are detected, remapped to QPSK symbols, OFDM modulated, and re-sent to the IoT terminal over the radio access link. Finally, the IoT terminal computes the LLRs of the QPSK symbols (see Section 2.4) and feeds them in the soft convolutional decoder to estimate the transmitted NB-IoT frames from the gateway (see Figure 3).

Let $s_{\mathrm{ntp}}\left[k\right]$ be the *k*-th NB-IoT modulated symbols, $g_{\mathrm{tx}}\left(\cdot \right)$ the transmit pulse-shaping square-root raised-cosine filter, $T_{s}$ the Nyquist symbol time of the NB-IoT signal, and δ the time-packing overlapping factor. Then, the time-packed signal that is generated by the gateway can be written as:(1)$$s_{\mathrm{tp}}\left(t\right)=\sum_{k}s_{\mathrm{ntp}}\left[k\right]\;g_{\mathrm{tx}}\left(t-k\left(1-\delta \right)T_{s}\right).$$

Next, the Electrical-to-Optical (E/O) conversion of $s_{\mathrm{tp}}\left(t\right)$ is done with a MZM driven by voltage:(2)$$v_{\mathrm{mzm}}\left(t\right)\;=\;V_{B}+\beta \;{\tilde{s}}_{\mathrm{tp}}\left(t\right)\;\left(V_{\pi }/\pi \right),$$
where $V_{B}$ and $V_{\pi }$ are the bias and half-wavelength voltages of the MZM, β is the intensity modulation index (scaling factor) that is selected for the communication in the optical feeder link, and:(3)$${\tilde{s}}_{\mathrm{tp}}\left(t\right)\;=\;s_{\mathrm{tp}}\left(t\right)/\sqrt{E\{|s_{\mathrm{tp}}\left(t\right){|}^{2}\}},$$
where E{·} is the mathematical expectation operator that determines the mean value of the signal.

After that, the time-packed signal is transmitted through the optical feeder link. At the satellite, the Optical-to-Electrical (O/E) conversion is performed with the aid of a Photodetector [28], obtaining:(4)$$r_{\mathrm{tp},\mathrm{sat}}\left[n\right]=\sqrt{{\left(\frac{E_{b}}{N_{0}}\right)}_{\mathrm{fl}}}\;h_{\mathrm{fl}}\;s_{\mathrm{tp}}\left[n\right]+\eta_{\mathrm{fl}}\left[n\right],$$
where $h_{\mathrm{fl}}$ is the equivalent channel gain of the optical feeder link (see Section 3), $s_{\mathrm{tp}}\left[n\right]$ is the *n*-th discrete sample of the time-packed signal in the gateway, and $\eta_{\mathrm{fl}}\left[n\right]$ is the resulting noise signal of unit power after the O/E conversion in the satellite (see Section 4.1). The value of ${\left(E_{b}/N_{0}\right)}_{\mathrm{fl}}$ represents the equivalent bit-to-noise-energy ratio of the optical feeder link.

The estimations of the non-time packed NB-IoT symbols at the satellite (i.e., ${\hat{s}}_{\mathrm{ntp},\mathrm{sat}}$) are obtained after the received signal samples in (Equation 4) are first matched-filtered and then equalized (see Section 2.3). Then, the 4-PAM non-time packed signals are re-mapped into QPSK symbols, OFDM modulated, and finally transmitted to the corresponding IoT device. Due to that, the *k*-th sample of the signal that the IoT device receives in the radio access link attains the form:(5)$$r_{\mathrm{ntp},\mathrm{iot}}\left[k\right]=\sqrt{{\left(\frac{E_{b}}{N_{0}}\right)}_{\mathrm{al}}}\;h_{a}\;{\hat{s}}_{\mathrm{ntp},\mathrm{sat}}\left[k\right]+\eta_{\mathrm{al}}\left[k\right],$$
where $\eta_{\mathrm{al}}\left[k\right]$ is the unit power noise signal, $h_{\mathrm{al}}$ is the gain of the radio access channel, and ${\left(E_{b}/N_{0}\right)}_{\mathrm{al}}$ is the bit-to-noise-energy ratio of the radio access link. Finally, the LLR of the received QPSK modulated symbols are computed, and soft-decoding it is conducted to detect the transmitted bits from the gateway, $b_{\mathrm{gw}}$ (see Figure 3).

#### 2.2.2. Case 2: Decode-and-Forward Relaying of NB-IoT Frames

Similar to *Case 1* (see Section 2.2.1), the optical receiver of the satellite removes the time-packed interference (see Section 2.3) but, instead of remapping the 4-PAM modulation to QPSK directly, it computes the LLRs of the 4-PAM modulation that are used by the soft-Viterbi decoder to detect the message bits transmitted from the gateway (see Section 2.4). After that, the estimated message bits ${\hat{b}}_{\mathrm{gw},\mathrm{sat}}$ are re-encoded, QPSK mapped, OFDM modulated, and forwarded to the corresponding IoT device. There, the NB-IoT receiver computes the LLRs of the receive QPSK symbols, to use them in the soft-Viterbi algorithm to detect the transmitted bits from the gateway ${\hat{b}}_{\mathrm{gw},\mathrm{iot}}$.

Let ${\hat{s}}_{\mathrm{ntp},\mathrm{sat}}\left[n\right]$ be the estimated symbols of the non-time-packed 4-PAM modulated symbols at the satellite. Then, the following step of *Case 2* consists in computing the two LLRs per modulated symbol of the 4-PAM constellation, denoted as $LLR_{b_{0}}$ and $LLR_{b_{1}}$ (see Section 2.3). After that, the soft-Viterbi decoder at the satellite estimates the original transmitted bits ${\hat{b}}_{\mathrm{gw},\mathrm{sat}}$, re-encodes them to regenerate the coded NB-IoT bits $c_{\mathrm{ntp},\mathrm{sat}}$, maps them to QPSK symbols $s_{\mathrm{ntp},\mathrm{sat}}$, and follows as in *Case 1* (see Figure 3).

#### 2.2.3. Case 3: Detect-and-Forward Relaying of NB-IoT Frames Encapsulated in DVB-S2(X)

The third case consists in encapsulating the NB-IoT encoded bits into DVB-S2(X) frames. Next, the LDPC coded bits from DVB-S2(X) are 4-PAM modulated, pulse-shaped, time-packed, and sent to the satellite. There, the time-packed interference is removed (see Section 2.3), and the LLRs for the 4-PAM modulation are computed and fed into the soft-LDPC decoder of the DVB-S2(X) [33]. After that, the NB-IoT encoded bits are des-encapsulated from the DVB-S2(X) frames, mapped to QPSK symbols, OFDM modulated, and forwarded to the IoT device. There, the LLRs of the QPSK symbols are first computed, and then used by its soft-Viterbi decoder to detect the message of the gateway.

Let $c_{\mathrm{iot},\mathrm{gw}}$ be the NB-IoT convolutional encoded signal at the terrestrial gateway, which is encapsulated into the DVB-S2(X) frame. Due to the difference in the payload size of a NB-IoT and a DVB-S2(X) frame, several NB-IoT convolutional codewords can be packed together into the input message of the DVB-S2(X) physical layer frames. We recall that at the physical layer, a DVB-S2(X) frame is first BCH encoded and then LDPC encoded. The BCH encoding is used to remove the possible error floors of the LDPC decoding, which aims at compensating the impairments that introduce the communication channel (i.e., the optical feeder link in our case). Let us assume that $c_{\mathrm{dvb},\mathrm{gw}}$ denotes the input frame to the DVB-S2(X) encoder at the gateway. Then, it can convey up to *P* NB-IoT convolutional-coded frames as $c_{\mathrm{dvb},\mathrm{gw}}=\left[c_{\mathrm{iot},\mathrm{gw}}\left(0\right)\cdots c_{\mathrm{iot},\mathrm{gw}}\left(P-1\right)\right]$. Next, the DVB-S2(X)-coded frames are 4-PAM modulated, E/O converted, and sent to the satellite on the optical feeder link. At the satellite, the O/E conversion is carried out, and the received signal samples are matched-filtered and equalized to mitigate the ISI introduced by time-packing. Likewise, in *Case 2*, the two LLRs are computed per 4-PAM symbol, i.e., $LLR_{b_{0}}$ and $LLR_{b_{1}}$ (see Section 2.3). However, in *Case 3* these LLRs are introduced into the soft LDPC decoder, not in the soft-Viterbi decoder as it was used in *Case 2*. After that, the NB-IoT encoded bits are de-encapsulated from the decoded DVB-S2(X) frames, QPSK mapped, OFDM modulated, and forwarded to the IoT terminal over the radio access link. Finally, at the IoT device, the processing for detecting the transmitted message is the same as the one that was explained for *Case 1*.

### 2.3. Equalization of the Time-Packed Signal

The three proposed relaying architectures rely on time-packing signalling in the optical feeder link to increase the throughput even further. Unfortunately, the use of time-packing introduces ISI that must be mitigated in reception at the satellite. The optimal strategy for cancelling this unwanted ISI consists in resorting the Maximum-Likelihood Sequence Decoding (MLSD) [34], which can be implemented efficiently with the Viterbi algorithm [35]. However, to increase the throughput of the optical feeder link, it is necessary use a low roll-off factor and large overlapping factor, two things that increase notably the complexity of the Viterbi Algorithm to be implemented in the satellite [27]. Due to the impracticability of this solution, alternative strategies should be considered. Towards this regard, in this paper we use a two-side Least Mean Square (LMS) filter to equalize the time-packed channel. Thus, the equalization strategy consists of two steps: (1) Compute the weights of the interference canceller by means of a training sequence and (2) apply the pre-computed weights to the received time-packed signal from the gateway. These weights have to be computed for each overlapping factor that can be used.

Let us assume that r is the vector that stacks the received samples of the time-packed signal, $w={\left[w[0]\cdots w[L-1]\right]}^{T}$ is the vector of *L* weights in the LMS equalizer, and y[n] is the buffer that contains the received samples that participates in the equalization process of the n-th time-packed received symbol. Then, the equalization of the n-th transmitted symbol attains the form:(6)$$z\left[n\right]=w^{T}\;y,\;y\;=\;{\left[r[n-(L-1)/2]\cdots r[n]\cdots r[n+(L-1)/2]\right]}^{T}.$$

During the training process, the ideal values of the signal samples at the output of the equalizer z[n] are known, enabling to determine the most convenient equalization weights. Specifically, the weights w have been computed by using the LMS algorithm strategy [26,36], i.e.,
(7)w[q]=w[q−1]+μe[q]y[q],
where w[p] contains the equalizer weights at the p-th training iteration, μ is the forgetting factor, and $e\left[p\right]=z_{t}\left[p\right]-z\left[p\right]$ is the error between the training symbol $z_{t}\left[p\right]$ and its estimation from (Equation 6).

### 2.4. Computation of the LLRs for the 4-PAM and the QPSK Modulation Schemes

In this paper, the NB-IoT transmitted data is encoded with a convolutional coding scheme. Furthermore, for Case 3, the encapsulated NB-IoT frames are protected using the LDPC code of the DVB-S2(X) standard. In both cases, soft-decoding is used and due to that, the LLRs has to be determined. However, the IoT devices and the satellite node may be limited in their power consumption. Fortunately, for both modulation schemes used in this paper (i.e., 4-PAM and QPSK), it is possible to derive reduced complexity closed-form formulas for computing the optimal LLRs.

The 4-PAM modulation scheme is used in the optical feeder link, whereas the QPSK modulation is used in the radio access link. We assume that Gray-mapping is used in both cases, and that their corresponding modulation symbols are $s_{4-\mathrm{pam}}=\left\{-3,-1,1,3\right\}/\sqrt{5}$ for 4-PAM and $s_{\mathrm{QPSK}}=\left\{-1-j,1-j,1+j,-1+j\right\}/\sqrt{2}$ for QPSK (see Figure 4). Since both modulation schemes transport two bits per modulated symbol, it is necessary to compute two LLRs per modulated symbol. Thus, the closed-form expression for computing these LLRs is given by:(8)$$LLR_{b_{m}}=-\log\left(\frac{\left(\sum_{p=0;\;s_{p}|b_{m}=0}^{M/2}e^{-\frac{|z-h\;s_{p}{|}^{2}}{2\sigma_{n}^{2}}}\right)}{\left(\sum_{q=0;\;s_{q}|b_{m}=1}^{M/2}e^{-\frac{|z-h\;s_{q}{|}^{2}}{(2\sigma_{n}^{2}}}\right)}\right),$$
where $LLR_{b_{m}}$ is the LLR that corresponds the *m*-th bit of the modulated symbol, $\sigma_{n}^{2}$ is the noise power, *M* is the number of constellation symbols, *h* is the communication channel, *z* is the received data at the input of the de-mapper, and $s_{p}$ ($s_{q}$) symbolizes the constellation symbols in which the *m*-th bit is 0 (1). Thus, according to (Equation 8) and the Gray mapping proposed in Figure 4, the LLR of the first bit $b_{0}$ for both 4-PAM and QPSK modulation schemes is given by:(9)$$LLR_{b_{0}}=-\log\left(\frac{\left(e^{-\frac{|z-h\;s_{0}{|}^{2}}{2\sigma_{n}^{2}}}+e^{-\frac{|z-h\;s_{1}{|}^{2}}{2\sigma_{n}^{2}}}\right)}{\left(e^{-\frac{|z-h\;s_{2}{|}^{2}}{2\sigma_{n}^{2}}}+e^{-\frac{|z-h\;s_{3}{|}^{2}}{2\sigma_{n}^{2}}}\right)}\right),$$
whereas LLR of the second bit $b_{1}$ attains the form:(10)$$LLR_{b_{0}}=-\log\left(\frac{\left(e^{-\frac{|z-h\;s_{0}{|}^{2}}{2\sigma_{n}^{2}}}+e^{-\frac{|z-h\;s_{3}{|}^{2}}{2\sigma_{n}^{2}}}\right)}{\left(e^{-\frac{|z-h\;s_{1}{|}^{2}}{2\sigma_{n}^{2}}}+e^{-\frac{|z-h\;s_{2}{|}^{2}}{2\sigma_{n}^{2}}}\right)}\right).$$

If (Equation 9) and (Equation 10) are computed for 4-PAM and QPSK modulation schemes, the closed-form expressions for $LLR_{b_{0}}$ and $LLR_{b_{1}}$ are shown in Table 1. In these closed-form formulas, we have that:(11)$$a=\left(2\;z\;s_{1}\right)/\left(2\;\sigma_{n}^{2}\right),\;b=\left(4\;s_{1}^{2}\right)/\left(2\sigma_{n}^{2}\right),\;c=3\cdot a,$$
where $s_{1}$ is the second symbol of the 4-PAM constellation, i.e., $s_{1}=-1/\sqrt{5}$ (see Figure 4) and signal *z* represents the data after equalizing the time-packed signal (see Section 2.3). For *Case 2*, this signal *z* corresponds to the NB-IoT one, and its LLRs are introduced in the soft-convolutional decoder for recovering the transmitted message to regenerate the NB-IoT signal. For *Case 3*, the signal *z* corresponds to the DVB-S2(X) after the time-packing equalization. The computed LLRs for this case is introduced to the LDPC decoder of the DVB-S2(X) receiver [33]. After that, the encoded NB-IoT bits are QPSK mapped, OFDM modulated, and forwarded to the IoT-device. Finally, at the IoT receiver, the LLRs of the received QPSK modulation symbols are computed for all cases under study. These LLRs are used by the soft-Viterbi decoder to recover the message transmitted from the satellite gateway.

## 3. Optical Wireless Satellite Channel Model

The optical signal that is used to transport the data symbols from the satellite gateway to the GEO satellite must go through the different layers of the Earth’s atmosphere. Unfortunately, the power loss that the optical feeder link experiences in the uplink direction of communication is larger than the one observed in downlink. This is because, in the ground-to-satellite communication, the optical signal starts to spread and accumulate distortion as soon as it leaves the satellite gateway transmitter.

### 3.1. Atmospheric Power Losses: Absorption and Scattering Modeling

The power loss of the optical feeder link is a function of the atmospheric attenuation, which depends on both absorption and scattering effects that the light signal experiences while propagating [37]. To compute this value, the atmospheric attenuation coefficient:(12)$$\gamma_{\mathrm{atm}}\;=\;\alpha_{m}+\alpha_{a}+\beta_{m}+\beta_{a}$$
needs to be computed, where $\alpha_{m}$ and $\alpha_{a}$ are the molecular and aerosol absorption coefficients, respectively, whereas $\beta_{m}$ and $\beta_{a}$ are the molecular and aerosol scattering coefficients, respectively.

**Modelling of absorption:** At Infrared (IR) wavelengths, the principal atmospheric absorbers are the molecules of water, carbon-dioxide, and ozone. As expected, the atmospheric absorption is a wavelength-dependent phenomenon. Therefore, the operating wavelength for optical feeder link transmissions should be chosen to minimize this loss, using the atmospheric transmission windows in which the molecular and aerosol absorption is less than 0.2 dB/km for clear sky conditions [38]. In addition to the low-absorption requirements, most optical feeder links are designed to work in the 780–850 nm and 1520–1600 nm windows because there are *off-the-shelf* lasers and detectors commercially available to work in these wavelengths.

**Modeling of scattering:** Like absorption, scattering is also a phenomenon that is strongly dependent on the operating wavelength. If the size of the atmospheric particles is small in comparison to the optical feeder link wavelength, then *Rayleigh scattering* is produced. Particles like air molecules and haze cause Rayleigh scattering [39] and affect notably optical wireless transmissions in the Visible Light (VL) and Ultraviolet (UV) regions; on the other hand, Rayleigh scattering can be neglected for optical feeder link wavelengths in the IR range (i.e., when λ≫1μm). Similarly, when the atmospheric particles size is comparable with the operating wavelength, then *Mie scattering* is produced. Aerosol particles, fog, and haze are the major contributors of Mie scattering, and this phenomenon is dominant for wavelengths in the IR range. Finally, if the atmospheric particles are much larger than the operating wavelength, the scattering is better described by geometrical optical models, which should be used in case of rain, snow, and hail weather conditions [40].

**Modeling the transmittance of the Earth’s atmosphere:** Apart from the previously described λ-dependent effects, the specific value that the atmospheric attenuation coefficient takes depends on the concentration of molecules (and aerosols) of the Earth’s atmosphere at different altitudes *h*. Based on this assumption, the atmospheric transmittance that an optical feeder link with zenith angle ζ experiences is given by:(13)$$T_{\mathrm{atm}}\left(\lambda \right)\;=\;\exp\{-\sec\left(\zeta \right)\int_{h_{0}}^{H}\gamma \left(\lambda ,h\right)\;dh\},$$
where $h_{0}$ is the altitude of the satellite gateway (ground station) over the sea level, *H* is the vertical height at which the GEO satellite is placed, and γ(λ,h) is the attenuation coefficient at wavelength λ and altitude *h*. Based on this formula, it is possible to see that atmospheric transmittance is increased at low zenith angles (i.e., at high elevation angles), as the fraction of the incident electromagnetic power that is transferred through the atmosphere layers is increased.

**Power losses due to fog:** From the common weather conditions, fog is the one that contributes most in the absorption and scattering of the optical signal when it propagates through the Earth’s atmosphere. In the presence of fog, the optical feeder link connectivity is seriously put at risk, particularly when the fog layer next to the ground station extends vertically very high, forming a fog layer that can be as thick as 400 m over the Earth’s surface. In such critical weather conditions, the use of very high power lasers (1550 nm) with special mitigation techniques is the only option to maximize the chances of optical feeder link connectivity. As an alternative method to the Mie scattering theory, the attenuation due to fog for different wavelengths can be estimated using empirical models that use as the input parameter the visibility in km measured on the VL region (550 nm). For a comparison of the fog attenuation at different wavelengths (850 nm and 950 mn), please refer to [41]. Note that in extreme cases, where the visibility due to fog is reduced to about 50 m, atmospheric attenuation can be as high as 350 dB/km [42].

**Power losses due to rain:** The impact of rain in the propagation of optical signals is not as pronounced as fog, because the size of the rain droplets are significantly larger in size (from 100μm to 1000μm) when compared to operating wavelengths of optical feeder links. For example, the attenuation loss in light rain (2.5 mm/h) to heavy rain (25 mm/h) ranges from 1 dB/km to 10 dB/km for 850 nm and 1550 nm operating wavelengths, respectively [43]. Note that the low clouds, which usually accompany rainy weather, are the source of strong attenuation in most optical feeder links. In order to combat the huge power loss that takes place in such conditions, it is recommended to include a few tens-of-dB margin (e.g., 30 dB) when designing the link budget of the optical wireless link. Moreover, optical feeder link designers can also implement adaptive coding and modulation schemes to address the varying weather conditions in the geographical area around the ground station [26].

**Power losses due to snow:** Finally, since the size of snow droplets is between the size of rain and fog droplets, the atmospheric attenuation for dry/wet snow conditions is usually stronger than the one in the presence of rain, but not as severe as the one in case of fog. However, during heavy snow storms, the path of the optical feeder link can be completely blocked for the presence of densely-packed snow flakes in the propagation path. In such cases, the attenuation is similar to the one observed in foggy weather (30–350 dB/km) and, as expected, can seriously put at risk the optical feeder link connectivity.

### 3.2. Atmospheric Turbulence: Beam Wander, Beam Spreading, and Beam Scintillation

Atmospheric turbulence is a random phenomenon that is caused by the variation of the temperature and pressure on the atmosphere layers that are in the propagation path of the optical wireless signal. These temperature and pressure inhomogeneities form turbulent cells, known as *eddies*, which have different sizes and different diffractive indexes. The eddies act as if prisms/lenses were deployed in the propagation path, introducing constructive and destructive interference in the received optical signal. The perturbations that atmospheric turbulence introduces in the wave-front of the optical beam can be physically described by the Kolmogorov model [44]. Depending on the size of the turbulent eddies with respect to the transmitted beam size, three types of atmospheric turbulence-induced effects can be identified, namely: *beam wander*, *beam spreading*, and *beam scintillation*.

**Turbulence-induced beam wander:** This phenomenon takes place when the size of the turbulent eddies is *larger* than the size of the optical beam. Beam wander results in a random deviation of the optical beam from its planned (rectilinear) propagating path and, in extreme displacement situations, may lead to the failure of the optical wireless link. Beam wander is a major concern in the uplink transmission of an optical feeder link, as the beam size in the ground-to-satellite transmission is often smaller than the size of the turbulent eddies, resulting in a beam displacement at the receiver side that can be as large as several hundred meters. 

In case of a collimated beam (plane wave model), the Root Mean Square (RMS) displacement due to beam wander for an uplink path with zenith angle ζ can be written as:(14)$$\begin{array}{ccc} \sigma_{\mathrm{BW}}^{2} & = & 7.25\;{\left(H-h_{0}\right)}^{2}\;\sec^{3}\left(\zeta \right)\;{W_{0}}^{-1/3}\;\int_{h_{0}}^{H}C_{n}^{2}\left(h\right){\left(1-\frac{h-h_{0}}{H-h_{0}}\right)}^{2}\;dh \end{array}$$(15)$$\begin{array}{ccc} & ≅ & 0.54{\left(H-h_{0}\right)}^{2}\sec^{2}\left(\zeta \right){\left(\frac{\lambda }{2W_{0}}\right)}^{2}{\left(\frac{2W_{0}}{r_{0}}\right)}^{5/3}, \end{array}$$
where *H* is the altitude of the GEO satellite (receiver), $h_{0}$ is the altitude of ground station (transmitter), $W_{0}$ is the initial beam size, and $r_{0}$ is the atmospheric coherence width, which is also known now as *Fried’s coherence length*, *Fried’s parameter*, or simply coherence length [45]. The Fried’s coherence length is a widely-used descriptor of the level of atmospheric turbulence at a particular site and, for a known structure constant profile $C_{n}^{2}\left(h\right)$ and plane wave model (collimated beam), it is given by: [46]
(16)$$r_{0}\;=\;{\left[0.423\;k^{2}\;\sec\left(\zeta \right)\int_{h_{0}}^{H}C_{n}^{2}\left(h\right)\;dh\right]}^{-3/5},$$
where k=2π/λ is the wavenumber of the optical beam. As expected, $C_{n}^{2}\left(h\right)$ varies with the time of the day, the geographical location, and the altitude. Therefore, for vertical optical links (slant paths), the value of $C_{n}^{2}\left(h\right)$ has to be integrated over the complete propagation path, starting from the height of the ground station above the sea level and ending at the altitude in which the Earth’s atmosphere vanishes (i.e., at about 40 km).

Various empirical models for $C_{n}^{2}\left(h\right)$ have been proposed in the literature to estimate the turbulence profiles, using as reference the experimental measurements that were carried out at different geographical locations, time of the day, wind speed, terrain types, among others. The most widely-used model to characterize the refractive index structure of the atmosphere for vertical links (slant paths) is the so-called Hufnagel-Valley (H-V) model [45], i.e.,
(17)$$C_{n}^{2}\left(h\right)\;\;=\;\;A_{0}\exp\left(\;-\;\frac{h}{100}\right)+5.94\times 10^{-53}{\left(\frac{v}{27}\right)}^{2}\;h^{10}\;\exp\left(\;-\;\frac{h}{1000}\right)+2.7\times 10^{-16}\;\exp\left(\;-\;\frac{h}{1500}\right),\;$$
where *h* [m] is the altitude, *v* [m/s] is the RMS wind-speed, and parameter $A_{0}\;=\;C_{n}^{2}\left(h_{0}\right)$ [m^−2/3^] is the nominal value of the refractive index near the ground level. The RMS wind speed in (Equation 17) is determined from the Bufton wind model, and can take values that range from v=10 to 30 m/s for moderate and strong wind speeds, respectively. Similarly, the ground turbulence level can take values between $A_{0}\;=\;1.7\times 10^{-14}$ and $1.7\times 10^{-13}$ m^−2/3^, which depends on the location and day time, among other parameters.

When using $A_{0}\;=\;1.7\times 10^{-14}$ m^−2/3^ and v=21 m/s, this model is commonly referred to as the H-V_5/7_ model because, for wavelength λ=0.5 μm and a transmitter on the ground looking up (i.e., with ζ=0 deg.), it predicts a value of atmospheric coherence diameter $r_{0}\;=\;5$ cm according to (Equation 16) and a value of an isoplanatic angle:(18)$$\theta_{0}\;=\;\frac{\cos^{8/5}\left(\zeta \right)}{{\left[2.91\;k^{2}\;\int_{h_{0}}^{H}C_{n}^{2}\left(h\right)\;{\left(h-h_{0}\right)}^{5/3}\;dh\right]}^{3/5}}$$
of 7μrad in case of a spherical wave with output-plane beam parameters Θ=Λ=0. The refractive index profile along the vertical/slant path is shown in Figure 5 for two nominal values of a refractive index at the ground level and three different RMS wind speeds. From this figure, it is possible to observe that the ground turbulence level $A_{0}$ has little effect above 1 km, and that the wind speed governs the profile behavior primarily in the vicinity of altitudes in the 10 km range.

Similarly, Figure 6 shows the RMS angular displacement due to beam wander ($\sigma_{\mathrm{BW}}^{2}$) as a function of the beam radius $W_{0}$, when operating wavelength λ=1.55μm and refractive index structure $C_{n}^{2}\left(h\right)$ follows the H-V_5/7_ model. As expected, the RMS beam wander displacement is higher for the largest zenith angle, as the section of the atmosphere through which the optical beam needs passes through is thicker, the beam deviation with respect to the straight path grows. Finally, according to (Equation 16), the Fried’s coherence length is $r_{0}\;=\;19.25$ and 12.70 cm for zenith angle ζ=0 and 60 deg., respectively.

**Turbulence-induced beam spreading:** This phenomenon takes place when the turbulent eddies are smaller than the size of the optical beam. Beam spreading generates a widening of the beam size, beyond the natural broadening due to diffraction that the non-turbulent atmosphere introduces. Beam spreading does not affect the direction of the optical beam but, in contrast, reduces the optical power at the receiver aperture due to the energy dispersion that takes place.

**Turbulence-induced beam scintillation:** When the size of turbulent eddies is of the *same order* of the size of the optical beam, then the eddies act as lenses that focus and defocus the incoming beam. In this situation, the eddies lead to a redistribution of the signal energy that generates a temporal and spatial fluctuation of the irradiance at the receiver aperture. This phenomenon, which is known as *scintillation*, represent one of the major sources of degradation in the performance of an optical feeder link. Atmospheric turbulence also leads to loss of spatial coherence of an initially coherent optical beam, and may also produce depolarization of the light and temporal stretching of the optical pulse. 

The atmospheric scintillation is measured in terms of the scintillation index, which is the normalized variance of the intensity fluctuations, i.e.,
(19)$$\sigma_{I}^{2}\overset{\Delta }{=}\frac{\langle {\left(I-I_{m}\right)}^{2}\rangle }{I_{m}^{2}}\;=\;\frac{\langle I^{2}\rangle -I_{m}^{2}}{I_{m}^{2}}=\frac{\langle I^{2}\rangle }{I_{m}^{2}}-1,\;I_{m}\;=\;\langle I\rangle ,$$
where *I* is the irradiance (intensity) in the detector plane and 〈·〉 denotes the ensemble average.

The Gamma-Gamma distribution has been proposed to describe the turbulence-induce scintillation over a broad range of beam diameters. The Probability Density Function (PDF) of the Gamma-Gamma turbulence model and the scintillation index are given by: (20)$$f_{I}\left(x\right)\;=\;\frac{2}{\Gamma (\alpha )\Gamma (\beta )x}{\left(\frac{\alpha \beta x}{I_{m}}\right)}^{\frac{\alpha +\beta }{2}}K_{\alpha -\beta }\left(2\sqrt{\frac{\alpha \beta x}{I_{m}}}\right)\;x\geq 0,\;\;\sigma_{I}^{2}\;=\;\frac{1}{\alpha }+\frac{1}{\beta }+\frac{1}{\alpha \beta },$$
respectively, where $I_{m}$ denotes the mean irradiance, Γ(x) is the Gamma function, and $K_{n}\left(x\right)$ is the modified Bessel function of the second kind. The parameters $\alpha \;=\;1/\sigma_{X}^{2}$ and $\beta \;=\;1/\sigma_{Y}^{2}$ of the Gamma-Gamma distribution in (Equation 20) are directly related to the atmospheric conditions, and for the untracked beam case are given: (21)$$\sigma_{X}^{2}\;=\;5.95{\left(H-h_{0}\right)}^{2}\;\sec^{2}\left(\zeta \right){\left(\frac{2W_{0}}{r_{0}}\right)}^{5/3}{\left(\frac{\alpha_{\mathrm{pe}}}{W}\right)}^{2}\exp\left\{\frac{0.49\;\sigma_{\mathrm{Bu}}^{2}}{{\left[1+\left(1+\Theta \right)\;0.56\;\sigma_{\mathrm{Bu}}^{12/5}\right]}^{7/6}}\right\}-1,$$
and
(22)$$\sigma_{Y}^{2}=\exp\left\{\frac{0.51\;\sigma_{\mathrm{Bu}}^{2}}{{\left[1+0.69\;\sigma_{\mathrm{Bu}}^{12/5}\right]}^{5/6}}\right\}-1.$$

The various parameters that appear in Equations (Equation 21) and (Equation 22) are defined as follows: (23)$$\alpha_{\mathrm{pe}}\;=\;\sigma_{\mathrm{pe}}/L,\;\sigma_{\mathrm{pe}}^{2}≅\sigma_{\mathrm{BW}}^{2}\left[1-{\left(\frac{C_{r}^{2}W_{0}^{2}/r_{0}^{2}}{1+C_{r}^{2}W_{0}^{2}/r_{0}^{2}}\right)}^{1/6}\right],\;L\;=\;\frac{H-h_{0}}{cos(\zeta )},\;C_{r}\;=\;2\pi ,$$
and are the jitter-induced angular pointing error, the pointing error variance, slant path length, and scaling constant, respectively. Similarly, the diffractive beam radius at the receiver is given by:(24)$$W\;=\;W_{0}\sqrt{\Theta_{0}^{2}+\Lambda_{0}^{2}},\;\mathrm{where}\;\Theta_{0}\;=\;1-\frac{L}{F_{0}},\;\Lambda_{0}\;=\;\frac{2L}{kW_{0}^{2}}$$
are the *input-plane beam parameters*. Note that for a collimated beam, the phase front radius of curvature at the transmitter output aperture $F_{0}\to \infty$ and, due to that, $\Theta_{0}≅1$. Finally, the irradiance flux variance in the focal plane of the receiver: (25)σBu2=8.7k7/6(H−h0)5/6sec11/6(ζ)×Re∫h0HCn2(h)ξ5/6Λξ+j(1−Θ¯ξ)5/6−Λ5/6ξ5/3dh,
where:(26)$$\xi \;=\;1-\frac{h-h_{0}}{H-h_{0}}$$
is the normalized distance for the uplink propagation path, and:(27)$$\bar{\Theta }\;=\;1-\Theta \;=\;1-\frac{\Theta_{0}}{\Theta_{0}^{2}+\Lambda_{0}^{2}}\;=\;-\frac{L}{F_{0}},\;\Lambda \;=\;\frac{\Lambda_{0}}{\Theta_{0}^{2}+\Lambda_{0}^{2}}\;=\;\frac{2L}{kW^{2}},$$
are the *output-plane beam parameters*.

In Figure 7 we plot the corresponding Gamma-Gamma PDF for three different beam sizes $W_{0}$, which are equal to 1, 10, and 50 cm. Once again, the wavelength was set to λ=1.55μm and the analysis was done for the uplink direction of communication of a perfectly vertical GEO satellite feeder link (i.e., ζ=0 deg. and *H* = 36,000 km) when using the H-V_5/7_ refractive index model (i.e., v=21 m/s). Note that in this situation, the Fried’s coherence length is $r_{0}\;=\;19.25$ cm. As expected, for small beam sizes in which the $2W_{0}/r_{0}\ll 1$ relationship is verified (e.g., similar to $W_{0}\;=\;1$ cm in Figure 7), the longitudinal component of the scintillation index will be much less than 1; due to that, the corresponding PDF of the normalized irradiance will have a shape that resembles the one of a log-normal distribution, but with some differences in the upper and lower tails. On the other hand, for large beams in which the $2W_{0}/r_{0}\gg 1$ relationship is observed (e.g., similar to $W_{0}\;=\;50$ cm in Figure 7), the scintillation index becomes larger than 1 and the shape of the PDF starts to resemble a negative exponential distribution.

## 4. Evaluation

The error correction capabilities that the Modulation and Coding Schemes (MCS) of the NB-IoT (and DVB-S2(X)) standard have on the end-to-end forward link of the GEO satellite system (i.e., from the satellite gateway to the IoT terminals) is now evaluated in detail. For this purpose, we first present the simulation setup and, after that, we show the different figures of merit that are relevant to characterize the end-to-end performance of the three GEO satellite relaying strategies.

### 4.1. Simulation Setup of the Optical Channel

According to the analysis presented in [28], the mean SNR of the electrical signal that is direct-detected by the PD that is placed on-board the satellite is given by
(28)$${\mathrm{SNR}}_{e,\mathrm{pd}}=\frac{E\{|i_{d}\left(t\right){|}^{2}\}}{E\{|n_{o}\left(t\right){|}^{2}\}}\approx \frac{I_{D}^{2}\;\beta^{2}}{E\{|n_{o}\left(t\right){|}^{2}\}}\;\beta \ll 1,$$
where
(29)$$I_{D}=E\left\{i_{D}\left(t\right)\right\}=\mu \;\frac{G_{o,\mathrm{tx}}\;G_{o,\mathrm{rx}}\;G_{o,\mathrm{edfa}}}{L_{o,\mathrm{fsl}}\;L_{o,\mathrm{atm}}\;L_{o,\mathrm{bsl}}\;L_{o,\mathrm{sys}}}\;\frac{P_{o,\mathrm{ld}}}{2}$$
is the DC component of the time-varying electrical current $i_{D}\left(t\right)$ that the PD generates when being excited by the intensity modulated optical signal, β is the intensity modulation index, and
(30)$$E\{|n_{o}{\left(t\right)|}^{2}\}\;=\;E\{|i_{\mathrm{shot}}{\left(t\right)|}^{2}\}\;+\;E\{|i_{\mathrm{thermal}}{\left(t\right)|}^{2}\}\;+\;E\{|i_{\mathrm{rin}}{\left(t\right)|}^{2}\}\;+\;E\{|i_{\mathrm{beat}}\left(t\right){|}^{2}\}\;$$
includes the contribution of all noise sources in the optical feeder link, namely the *shot noise* sources, *thermal noise*, *Relative Intensity Noise* (RIN) of LD, and *beat noise* [25]. Note that shot noise term includes the contribution of the received optical signal, the Amplified Spontaneous Emission (ASE) noise, the background optical noise and the dark current noise, whereas the beat noise term accounts the effect of combining the received optical signal with the ASE noise.

When the received optical power is between −90 and −20 dBW, it can be shown that the beat noise between received optical signal and ASE noise dominates the SNR of the optical feeder link [47]. In this situation,
(31)$$E\{|n_{o}{\left(t\right)|}^{2}\}\;\approx \;E\{|i_{\mathrm{beat}}\left(t\right){|}^{2}\}\;=\;i_{\mathrm{sig}-\mathrm{sp}}^{2}+i_{\mathrm{sp}-\mathrm{sp}}^{2}\approx i_{\mathrm{sig}-\mathrm{sp}}^{2}\;=\;4\;I_{D}\;I_{\mathrm{ase}}\left(B_{e}/B_{o}\right),$$
where $B_{o}$ is the bandwidth of the optical signal at the PD input, $B_{e}$ is the bandwidth of the electrical signal at the PD output, and $I_{\mathrm{ase}}=\mu \;G_{o,\mathrm{edfa}}\;P_{\mathrm{ase}}$ is the DC component generated by the ASE noise, whose equivalent noise power at the input of the EDFA is given by $P_{\mathrm{ase}}=\rho_{\mathrm{ase}}\;B_{o}$.

Table 2 summarizes the parameters of the optical feeder link, taking into account both the optical gains and losses, as well as the different sources of optical noise [25,48]. Note that the values of the parameters that appear in this table have been determined taking into account the different phenomena described in Section 3, when modeling the contribution of each of them in the space-to-ground optical link budget. The effect of any other parameter not listed in this table is considered as negligible. When $L_{o,\mathrm{atm}}=0\;\mathrm{dB}$ (i.e., clear-sky conditions), the DC current at the PD output is $I_{D}=75.68$ mA, whereas the DC current generated by the ASE noise is $I_{\mathrm{ase}}=0.125$ mA regardless of the weather. Thus, when we set the intensity modulation index β=0.5, the SNR of the electrical signal at the PD output becomes
(32)$${\mathrm{SNR}}_{e,\mathrm{pd}}\left[\mathrm{dB}\right]=25.01\left[\mathrm{dB}\right]-L_{o,\mathrm{atm}}\left[\mathrm{dB}\right].$$
Note that for larger intensity modulation indexes β, the non-linear distortion introduced by the MZM of the optical transmitter becomes more notable. For those situation, the use of Digital Pre-Distortion (DPD) is necessary to keep non-linear distortion under control [25].

Figure 8 shows the Cumulative Distribution Function (CDF) of the power loss that turbulence-induced beam wander and beam scintillation introduce on the received optical power of the optical uplink transmission in the satellite forward link (see Section 3). Without loss of generalization, it is assumed that the ground station is placed next to the city of Madrid, Spain (i.e., Lat. 40.43° North, Long. 4.25° West, and altitude $h_{0}=864$ m), with an expected system availability due to cloud-free line-of-sight conditions close to 99.8%. Similarly, the satellite is assumed to be placed in the GEO position of 19.2° East at an altitude of *H* = 36,000 km, giving as result a zenith angle ζ=52.61 deg. for the optical feeder link in the forward direction of communication. The nominal value of the refractive index near the ground level is $A_{0}=1.7\times 10^{-14}$ m^−2/3^, and the CDF is plotted for three different wind speeds (i.e., v = 10, 21 and 30 m/s). Note that when v=21 m/s, the refractive index model becomes the well-known H-V_5/7_ model. Finally, an untracked collimated beam is assumed, with a beam radius of $W_{0}=10$ cm at a wavelength of λ=1550 nm. As expected, the stronger is the wind speed, the more variability is introduced in the instantaneous intensity of the received optical signal at the GEO satellite. For example, the 10-th percentile for the turbulence-induced power loss (i.e., power loss margin for a 90% link availability without coding) is equal to 2.4, 3.0 and 3.8 dB for a wind speed of 10, 21 and 30 m/s, respectively. These turbulence-induced power losses grow to 4.3, 5.4 and 7.0 dB when we study the 1-st percentile (i.e., power loss margin for a 99% link availability without coding) for the same three values of wind speeds (i.e., 10, 21 and 30 m/s, respectively).

### 4.2. Simulation Setup of NB-IoT Signal

This paper evaluates the BLER and Throughput of the following three forward link architectures for transmitting NB-IoT over satellite when the feeder link is optical:Case 1: NB-IoT detect-and-forward (See Section 2.2.1).Case 2: NB-IoT decode-and-forward (See Section 2.2.2).Case 3: NB-IoT encapsulated in DVB-S2(X) frames. (See Section 2.2.3).

In all cases, the NB-IoT frame length is set to N=1032 bits. The pulse-shaping filter for NB-IoT and DVB-S2(X) waveforms is assumed to be Square-Root Raised-Cosine (SRRC) with roll-off factor ρ=0.15. The overlapping factors that have been considered in the simulations to implement the time-packing signaling are δ=0, 15, 25 and 40%. Note that when δ=0% (i.e., no overlapping), the symbol time is equal to the *Nyquist* symbol time. The modulation scheme used in the optical feeder links is 4-PAM, in order to make it compatible with the Intensity Modulation/Direct Detection approach. On the contrary, the access link uses QPSK, which is one of the modulation schemes considered for the downlink in the NB-IoT standard. So, in both optical feeder and radio access links, the number of modulation states is M=4 and the number of bits per modulated symbol is $N_{b}=2$.

In order to make fair comparisons, all the relaying satellite architectures under analysis use the same data rate. In particular, the BLER and throughput of the different satellite relaying strategies have been tested with nine possible code rates, namely: Three *low* code rates that are more robust than the mother code rate of the convolutional encoding (i.e., 0.2222, 0.259 and 0.3), three *medium* code rates that provide similar error correction capabilities than the mother code rate (i.e., 0.4444, 0.5319 and 0.6), and three *high* code rates that are the least robust (i.e., 0.6386, 0.7678 and 0.854). For the first and second use cases, these code rates correspond to the code rate of the NB-IoT waveform. However, for the third case, these code rates represent the joint code rate of the DVB-S2(X) and NB-IoT waveforms. This joint code rate is defined as the product between the code rates of the DVB-S2(X) and NB-IoT signals as
(33)$$R_{c,\mathrm{case}\;3}=R_{c,\mathrm{iot}}\times R_{c,\mathrm{dvb}}.$$
In (Equation 33), the code rates of NB-IoT and DVB-S2(X) have been selected such that the their joint code rate $R_{c,\mathrm{case}\;3}$ is equal to the code rates of NB-IoT for the first and second cases. Thus, three code rates (i.e., low, medium and high) have been considered for each waveform (i.e., NB-IoT and DVB-S2(X)), which enable to obtain the nine code rates for cases 1 and 2. Specifically, the code rates of the DVB-S2(X) and NB-IoT standards that have been used are: $R_{c,\mathrm{dvb}}=0.66667$, 0.8 and 0.9 [33] and $R_{c,\mathrm{iot}}=0.3333$, 0.66667 and 0.96, respectively. Thus, we obtain nine possible combinations of the code rates, which are values that are similar to the ones used in the three relaying architectures (see Table 3 for more details).

Therefore, after obtaining a similar code rate for the three cases under study, the architecture that provides the best in terms of BLER and throughput is determined for the different SNR working regimes in the optical feeder link (uplink) and radio access link (downlink), respectively. Regarding the throughput, the following closed form formula has been used to estimate its value, i.e.,
(34)$$\mathrm{THR}\left(\delta ,R_{c}\right)=\frac{N_{b}\;R_{c}\;\left(1-\mathrm{BLER}\right)}{\left(1+\rho )\;(1-\delta \right)},$$
where the code rate $R_{c}$ for the two first cases under study corresponds to the code rate of the NB-IoT waveform. For the third case, the value of $R_{c}$ represents the joint code rate of the NB-IoT and DVB-S2(X) waveforms, which is given in (Equation 33). Without loss of generality, the roll-off factor is kept fixed in all simulations (i.e., ρ=0.15). Thus, according to the overlapping factors and code rates that have been previously presented, the maximum achievable throughput in each situation are shown in Table 4.

This maximum achievable throughput may not be reached due to the presence of a residual time-packing interference, unexpected variations in the optical channel gain, and noise in the optical feeder link may increase the BLER. So, simulation results give information about the transmission structure that provides the best in terms of both BLER and throughput. Note that lowest code rates offer a very low BLER but, in contrast, worst results in terms of throughput. On the contrary, high code rates may provide the highest achievable throughput but, at the same time, the worst BLER. For this reason, a target $\mathrm{BLER}\leq 10^{-2}$ has been considered as requirement for a normal service according to what is stated in the 5G specifications. This condition helps to decide the best transmission architecture, such that the case that satisfies the target BLER and provides the best throughput will be categorized as the most convenient transmission architecture. However, in order to compare the three cases under study, additional figures of merit needs to be defined in the following section.

### 4.3. Definition of the Figures of Merit

In the previous section, we defined the throughput in terms of the overlapping factor and code rate. However, in the evaluation process of the three cases, it is important to identify the envelope of the throughput defined in (Equation 34). Specifically, we will study the envelope of the throughput when the overlapping factor δ is kept constant. This throughput, denoted as $\mathrm{THR}{\left(\delta \right)}^{*}$, is defined as the maximum throughput for all code rates with the same overlapping factor, and is computed as follows:(35)$$\mathrm{THR}{\left(\delta \right)}^{*}=\max_{R_{c}}\left\{\mathrm{THR}(\delta ,R_{c})\right\}.$$

Next, it is also interesting to determine the envelope of the throughput for the no-time-packing and time-packing signaling schemes. For the case of no-time-packing signaling, the envelope of its throughput is obtained by setting δ=0 in (Equation 35), that is, ${\mathrm{THR}}_{\mathrm{ntp}}=\mathrm{THR}{\left(0\right)}^{*}$. For the case of time-packing signaling, the envelope of the throughput is determined by computing the maximum of (Equation 35) for all overlapping factors that are higher than zero (i.e., δ > 0), according to
(36)$${\mathrm{THR}}_{\mathrm{tp}}=\max_{\delta >0}\left\{\mathrm{THR}{\left(\delta \right)}^{*}\right\}.$$

Then, the overall envelope of the throughput for a particular relaying strategy is obtained from the maximum envelope throughput of the time-packed and non-time packed ones, i.e.,
(37)$${\mathrm{THR}}_{\mathrm{case}\;q}=\max\left\{{\mathrm{THR}}_{\mathrm{ntp}},{\mathrm{THR}}_{\mathrm{tp}}\right\}\;q=1,2,3.$$

After defining the throughput envelope in different working conditions, we now introduce two relative figures of merit that evaluate the *throughput gain* of the relaying strategies. The first one determines whether using time-packing is beneficial or not with respect to non-time-packing. This parameter, which provides the percentage gain in throughput, attains the form
(38)$$G_{\mathrm{thr}}\left(\%\right)=\frac{\left({\mathrm{THR}}_{\mathrm{tp}}-{\mathrm{THR}}_{\mathrm{ntp}}\right)\times 100}{{\mathrm{THR}}_{\mathrm{ntp}}}.$$

The second figure of merit aims at determining the relative gain in throughput of the different relaying strategy. From the three cases under analysis, the first one represents the simplest approach, since the satellite does not manipulate the content of the NB-IoT frame beyond the symbol-by-symbol mapping that is needed to adapt the optical wireless transmission (*M*-PAM) to radio wireless (*M*-QAM). Therefore, it is considered as the benchmark system. Then, the relative throughput gain that cases 2 and 3 provide with respect to case 1 is given by
(39)$$\Delta G_{\mathrm{thr},(q,1)}\left(\%\right)=\frac{\left({\mathrm{THR}}_{\mathrm{case}\;q}-{\mathrm{THR}}_{\mathrm{case}\;1}\right)\times 100}{{\mathrm{THR}}_{\mathrm{case}\;1}}\;q=2,3,$$
where ${\mathrm{THR}}_{\mathrm{case}\;q}$ is the throughput envelope for the three relaying strategies under study, computed with the aid of (Equation 37). Finally, after defining the figures of merit to carry out the analysis, we are ready to present the obtained simulation results.

### 4.4. Simulation Results

The results provided in this section evaluate the BLER and throughput of the three aforementioned NB-IoT satellite relaying architectures in the following aspects:Evaluation of the end-to-end BLER and throughput when the $E_{b}$/$N_{0}$ of the feeder link varies from 2 to 22 dB in steps of 1 dB (see Figure 9), the SNR of the access link is 20 dB, the wind speed is 21 m/s, and the NB-IoT code rates are $R_{c,\mathrm{iot}}=0.3333$, 0.66667, and 0.95556. There is no time-packing (δ=0%) and configuration is Case 1 (Detect-and-Forward NB-IoT). The throughput of Figure 9 is the $\mathrm{THR}\left(0,R_{c,\mathrm{iot}}\right)$ of (Equation 34);Evaluation of the end-to-end BLER and throughput when the $E_{b}$/$N_{0}$ of the access link varies from −8 to 12 dB in steps of 1 dB (see Figure 10), the SNR of the feeder link is 15, 20, and 25 dB, the wind speed is v=10, 21, and 30 m/s, and the NB-IoT code rates are $R_{c,\mathrm{iot}}=0.3333$, 0.66667 and 0.95556. There is no time-packing (δ=0%) and the tested configuration is Case 1 (Detect-and-Forward NB-IoT). The throughput of Figure 10 is $\mathrm{THR}\left(0,R_{c,\mathrm{iot}}\right)$ of (Equation 34);Evaluation of the throughput for Case 1 (see Figure 11): Detect-and-Forward NB-IoT when the SNR of the feeder link is 15 dB, the $E_{b}/N_{0}$ of the access varies from −8 to 12 dB, and the code rates of the NB-IoT and overlapping factors are as defined in Section 4.2. The throughput of Figure 11 corresponds to $\mathrm{THR}\left(\delta ,R_{c,\mathrm{iot}}\right)$ of (Equation 34);Evaluation of the throughput for Case 2 (see Figure 12): Decode-and-Forward NB-IoT when the SNR of the feeder link is 15 dB, the SNR of the access varies from 0 to 20 dB, and the code rates of the NB-IoT and overlapping factors are as defined in Section 4.2. The throughput of Figure 12 corresponds to $\mathrm{THR}\left(\delta ,R_{c,\mathrm{iot}}\right)$ of (Equation 34);Evaluation of the throughput for Case 3 (see Figure 13): Detect-and-Forward NB-IoT encapsulated in DVB-S2(X) frames when the SNR of the feeder link is 15 dB, the SNR of the access varies from 0 to 20 dB, and the code rates of the DVB-S2(X) and NB-IoT and overlapping factors are defined as in Section 4.2. The throughput of Figure 13 corresponds to $\mathrm{THR}\left(\delta ,R_{c,\mathrm{iot}}\right)$ of (Equation 34);Evaluation of the envelope of the throughput when the overlapping factor δ remains constant for all cases under study (see Figure 14a–c) when the SNR of the feeder link is 15 dB, the SNR of the access varies from 0 to 20 dB, and the code rates of the DVB-S2(X)/NB-IoT frames and the overlapping factors are defined in Section 4.2. The throughput of these figures corresponds to $\mathrm{THR}{\left(\delta \right)}^{*}$ of (Equation 35);Evaluation of the gain in throughput of time-packing signaling with respect to no-time-packing, as shown in Figure 14d. The SNR of feeder link is 15 dB, the SNR of the access varies from 0 to 20 dB. This gain is denoted as $G_{\mathrm{thr}}\left(\%\right)$ and is computed following (Equation 38);Comparison among the envelope of the throughput for all cases under study (see Figure 15a–d) in terms of the overlapping factor δ, when the SNR of feeder link is 15 dB, the SNR of the access link varies from 0 to 20 dB, and the code rates of the DVB-S2(X)/NB-IoT frames and overlapping factors are defined as in Section 4.2. The throughput of these figures corresponds to $\mathrm{THR}{\left(\delta \right)}^{*}$ of (Equation 35);Comparison of the envelope of the throughput for all cases under study (see Figure 16a) when the SNR of the feeder link is 15 dB, the SNR of the access link varies from 0 to 20 dB. The throughput plotted in this figure is denoted as ${\mathrm{THR}}_{\mathrm{case}\;q}$ and is computed according to (Equation 37);Comparison of the gain in throughput offered by Cases 2 and 3 with respect to Case 1 (see Figure 16b) when the SNR of the feeder link is 15 dB, the SNR of the access link varies from 0 to 20 dB. The gain in throughput of this figure is denoted as $\Delta G_{\mathrm{thr},(q,1)}\left(\%\right)$ with q=2,3, and is determined following (Equation 39).Evaluation of the BLER for Case 1 (Detect-and-Forward NB-IoT), Case 2 (Decode-and-Forward with NB-IoT) and Case 3 (Decode-and-Forward with NB-IoT/DVB-S2(X)), whose performance curves can be found in Figure 17, Figure 18 and Figure 19, respectively.

From the simulations of the end-to-end BLER and throughput in terms of the SNR in the optical feeder link (uplink), assuming a very high SNR in the radio access link (see Figure 9), it is possible to observe the region in which the uplink communication limits the performance of the end-to-end transmission. In all these plots, the dotted lines represents the cases where the BLER is higher than the target BLER (i.e., BLER>0.01), whereas the continuous lines identify the situations in which the BLER is lower than the target BLER (i.e., BLER≤0.01). For a better indication, the grey area in the figure shows the region of SNRs in which the tested code rates for NB-IoT satisfies the target BLER. Thus, it is possible to observe that $R_{c,\mathrm{iot}}=0.3333$ provides the best throughput when the SNR of the feeder link is ranged between 13 and 19 dB, $R_{c,\mathrm{iot}}=0.6666$ is the best option when the feeder link SNR is between 19 and 22 dB, and finally $R_{c,\mathrm{iot}}=0.95556$ is the most convenient alternative when the SNR in the uplink direction of communication is larger than 22 dB. According to these results, we now consider that the SNR of the feeder link is 15, 20, and 25 dB, for determining the degradation that different wind speeds introduce on the end-to-end link.

Figure 10 evaluates the degradation of the BLER (figures on the left) and throughput (figures on the right) when the wind speed is v=10, 21 and 30 m/s, and the SNR in the uplink direction of communication is 15, 20 and 25 dB. According to these results, the higher is the SNR of the optical feeder link, the lower is the degradation that the wind introduces on the end-to-end BLER and throughput of the forward NB-IoT link. Consequently, a SNR of 15 dB is used as reference value in the optical feeder link from now on, to compare the performance of the three satellite relaying architectures under study.

Figure 11 presents the throughput for each overlapping factor and code rate for Case 1 (Detect-and-forward with NB-IoT). Simulation results show that the segment of the NB-IoT code rates that satisfies the target BLER is the lowest one (i.e., $R_{c,\mathrm{iot}}=0.2222$, 0.259, and 0.3) for all the overlapping factors under study. The medium code rate of $R_{c,\mathrm{iot}}=0.4444$ could only achieve its maximum throughput with no overlapping (i.e., when δ=0%). Medium and larger code rates are mainly penalized by the SNR of the optical feeder link (uplink), and they are not able to achieve their maximum throughput (see Table 4). When comparing time-packing versus no-time-packing, results show that time-packing signaling provided a better throughput, and the gain was depended on the $E_{b}/N_{0}$ of the access link.

Figure 12 shows the throughput for each code rate and overlapping factor for Case 2 (Decode-and-forward with NB-IoT). From this figure, it is possible to observe that the regeneration of the NB-IoT signal in the satellite permits to use also medium code rates (i.e., $R_{c,\mathrm{iot}}=0.4444$) with overlapping factor δ=0 and 15%. Therefore, it means that the largest throughput is not obtained with the largest possible overlapping factor (i.e., δ=40%) and low code rates (i.e., $R_{c,\mathrm{iot}}=0.3$) as in the previous case, but with a low-to-moderate overlapping factor (i.e., δ=15%) and a medium code rate (i.e., $R_{c,\mathrm{iot}}=0.4444$).

Similarly, Figure 13 shows the throughput for each code rate and overlapping factor for Case 3 (Decode-and-forward with NB-IoT and DVB-S2(X)). Results of this figure show that the best code rate to use in the optical feeder link is $R_{c,\mathrm{dvb}}=0.6666$. For the NB-IoT system, in contrast, the best option is $R_{c,\mathrm{iot}}=0.3333$ when the $E_{b}$/$N_{0}$ in the access link varies between −1 and 5 dB, $R_{c,\mathrm{iot}}=0.6666$ when the $E_{b}$/$N_{0}$ in the access link grows from 5 to 9 dB, and finally $R_{c,\mathrm{iot}}=0.96$ for $E_{b}/N_{0}$ values beyond 9 dB in the access link.

Figure 14a–c show the envelope of the throughput for Cases 1–3 and the different overlapping factors under study. From these figures, it is possible to observe that for Cases 1 and 3, the larger is the overlapping factor, the higher is the maximum throughput that is achieved when δ=40%. On the contrary, for Case 2, the maximum throughput is achieved when the overlapping factor is δ=15%. Figure 14d shows the percentage gain in throughput of the time-packed versus no-time-packed signalling. From this sub-figure, it is possible observe that for when the access link $E_{b}/N_{0}$ is between −2 and 5 dB, the percentage gain of Case 1 is in the order of 50%. For other values of $E_{b}/N_{0}$, this gain falls down to about 10%. For Case 2, the percentage gain on the throughput envelope of time-packed schemes is larger than 40% when the $E_{b}/N_{0}$ varies between −4 and 1 dB. However, for values of $E_{b}/N_{0}$ larger than 1 dB in the access link, this gain in the throughput envelope reduces to about 20%. For Case 3, we have that for an $E_{b}/N_{0}$ larger than 0 dB, the minimum gain in the throughput envelope is 40%. Though Case 3 provides the largest gain, the minimum $E_{b}/N_{0}$ in the access link to use time-packing is larger than in Cases 1 and 2.

Figure 15 shows the envelope of the throughput for all cases in terms of the overlapping factor. From these figures, it is possible to observe that for Cases 1 and 3, the larger is the overlapping factor, the larger is the maximum throughput that is feasible when δ=40%. On the contrary, for Case 2, the maximum throughput is achieved when δ=15%.

Next, Figure 16 presents the envelope of the throughput for all cases (Figure 16a) and the percentage gain of Cases 2 and 3 with respect to Case 1 (Figure 16b). Studying Figure 16a, it is possible to conclude that Cases 2 and 3 have a better throughput than Case 1 in most cases. Thus, Case 1 is used as benchmark to assess the goodness of Cases 2 and 3. From this figure, it can also be noted that the regeneration of the NB-IoT signal enables to lower the minimum $E_{b}/N_{0}$ to about 1.5 and 4 dB for Case 2 and Case 3, respectively. Regarding the gain in throughput for Cases 2 and 3 with respect to Case 1, for $E_{b}/N_{0}$ values lower than −2 dB, the best option is to use Case 2; for an $E_{b}/N_{0}$ ranging between −2 and 4 dB, both Cases 2 and 3 provide practically the same throughput. Finally, for $E_{b}/N_{0}$ larger than 4 dB, the best strategy is to encapsulate the NB-IoT frames into the DVB-S2(X) ones (Case 3). Finally, Figure 17, Figure 18 and Figure 19 show the BLER of the three relay cases under study.

## 5. Conclusions and Futures Extensions

This section presents the main conclusions of the paper and possible future research to extend this work.

### 5.1. Concluding Remarks

Future mobile systems over satellite may suffer a bottleneck in the feeder link when they have to provide connectivity to OTA applications on a global scale, such as software and firmware updates for autonomous driving. To tackle this issue, the use of optical wireless technologies has been seriously considered to implement the feeder link and satisfy the high data rate demand that will be required. In this regard, this paper presented closed-form formulas that estimate the impairments that are introduced in the optical uplink transmission in terms of the wind speed, wavelength of the optical signal, beam width and telescope aperture, refractive index structure of the atmosphere, and azimuth angle of transmission, among others. Furthermore, the use of time-packing has been also considered to increase the spectral efficiency of the optical feeder link even further. Three different relaying strategies have been analyzed for the GEO satellite, namely: (1) Detect-and-Forward with NB-IoT; (2) Decode-and-Forward with NB-IoT; and (3) Decode-and-Forward with NB-IoT/DVB-S2(X). From the simulation results that were obtained, it was possible to conclude that the wind speed has little effect on the end-to-end for mean SNR values larger than 20 dB in the optical feeder link. For this reason, the performance of the different relaying strategies has been studied in further detail for an optical feeder link at 15 dB (i.e., up to 10 dB atmospheric loss in the optical link budget).

For fair comparisons among all the cases under study, a similar code rate has been included. For Detect-and-Forward and Decode-and-Forward relaying architectures with NB-IoT, these code rates correspond to the ones defined in the NB-IoT standard. In contrast, for Decode-and-Forward with NB-IoT/DVB-S2(X), the equivalent code rate is equal to the product between the code rates of the DVB-S2(X) and NB-IoT frames, which can be obtained by using two MCS with high code rates. After evaluating these relaying configurations, it was possible to conclude that the Detect-and-Forward with NB-IoT achieved its larger throughput when using the lowest NB-IoT code rate and the largest overlapping factor. In contrast, the relaying architecture based on Decode-and-Forward with NB-IoT obtained its largest throughput when using intermediate code rates and intermediate overlapping factors. Finally, for the Decode-and-Forward with NB-IoT/DVB-S2(X) relaying architecture, the largest throughput was obtained when using an intermediate code rate for DVB-S2(X), with a variable code rate for NB-IoT that was adjusted according to the $E_{b}/N_{0}$ in the radio access links. In this latter case, the larger the $E_{b}/N_{0}$, the higher the code rate that NB-IoT should be used. Regarding the most convenient overlapping factor, the highest throughput was observed when using the largest overlapping factors.

All the proposed relaying architectures were evaluated assuming time-packed signalling in the optical feeder link. The ISI introduced by time-packing was mitigated using an adaptive MMSE equalizer. From simulations, it was concluded that the use of time-packing increases the throughput in all proposed regenerative strategies under study, when compared to the cases in which time-packing was not used. However, this gain was depended on the $E_{b}/N_{0}$ of the access link. For Detect-and-Forward with NB-IoT, this gain varied from 10% to 50–65% for low and medium $E_{b}/N_{0}$ in the access link, respectively. For Decode-and-Forward with NB-IoT, the gain in throughput observed with time-packed signalling with respect to no-time packed going from 45–65% for low $E_{b}/N_{0}$ of the access link to 10% for medium-to-large $E_{b}/N_{0}$ values. Therefore, the decoding of the information permits to achieve higher gains at lower $E_{b}/N_{0}$. However, at larger $E_{b}/N_{0}$, the behavior of time-packed and no-time-packed signals is quite high and, as consequence, the improvement in throughput when comparing both schemes is reduced. Finally, for the Decode-and-Forward with NB-IoT/DVB-S2(X), the gain in throughput varies between 50–65% for medium $E_{b}/N_{0}$ values in the access link to 45% for high $E_{b}/N_{0}$ ones. In this scenario, the large error correction capability of LDPC codes enables one to remove the residual ISI that the equalizer is not able to eliminate. Consequently, it can deal successfully with the strong ISI that time-packing with large overlapping factors introduce.

Next, when comparing the aggregate throughput that the three relaying strategies offer, it can be concluded that the ones based on the Decode-and-Forward architecture provide the largest values. Specifically, the Decode-and-Forward with NB-IoT offers a better throughput for low $E_{b}/N_{0}$ in the access link, whereas the Decode-and-Forward with NB-IoT/DVB-S2(X) provides a larger throughput for medium-to-high $E_{b}/N_{0}$ values. Note that Decode-and-Forward with NB-IoT regenerates the NB-IoT signal in the satellite, whereas the Decode-and-Forward with NB-IoT/DVB-S2(X) protects the NB-IoT symbols by introducing the code rate of DVB-S2(X) in the optical feeder link. In this situation, when the $E_{b}/N_{0}$ of the access link was low, the LDPC decoder of DVB-S2(X) did not converge and, due to that, it introduced errors in the NB-IoT frames that were decoded on-board the satellite for re-transmission in the radio access link. In this situation, these NB-IoT frames were protected with a code rate larger than their equivalent for the other cases; therefore, it can be concluded that at a low $E_{b}/N_{0}$ in the radio access link, the Case 3 relaying strategy is not the optimum one. However, when the $E_{b}/N_{0}$ of the access increased, the LDPC decoder started to converge and, due to that, it managed to remove the erroneous bits that were introduced by the optical feeder link in the NB-IoT frames that were encapsulated in the DVB-S2(X) signal. Here, as the NB-IoT frames had a higher code rate than the code rates that were used for other configurations, it offered the largest throughput.

Finally, it is remarked that the link layer should be prepared for adjusting dynamically the transmission architecture according to the $E_{b}/N_{0}$ of the access link. Therefore, a control channel from the NB-IoT terminals to the gateway would be necessary, such that the estimated $E_{b}/N_{0}$ of the access link can be known in advance when selecting the MCS of the NB-IoT frame, enabling to improve the end-to-end throughput. In all cases, the IoT terminal would receive the data in NB-IoT signalling format. Note that all required modifications of the Decode-and-Forward of NB-IoT with and without DVB-S2(x) encoding would be transparent to the IoT terminal, since the time-packing signalling, NB-IoT regeneration, and the DVB-S2(X) encapsulation would be performed at the satellite feeder link.

### 5.2. Future Extensions

After presenting the main conclusions, we introduce possible future research extensions of this work. Specifically, the following ones are considered: (i) Adapt the optical channel to LEO and MEO scenarios, in order to evaluate the performance of the proposed relaying strategies at different satellite orbits; (ii) evaluate the benefits of using time-packing with optical feeder link in the reverse flow of information, from the IoT terminals to the satellite gateway; and (iii) study the viability of using time-packing techniques as a potential physical layer security scheme. In the first case, the aim would be to assess the benefits that LEO and MEO satellites could reap from the proposed relaying architectures. For both LEOs and MEOs, the slant range and elevation angle of the feeder link changes continuously with the time and, as consequence, the optical channel modeling has to be adjusted accordingly to consider these effects. The complexity of the proposed schemes should be also analyzed in detail, since LEO and MEO satellites have more energy constraints than GEO ones due to the Earth/Moon possibly blocking the sunlight that reaches their solar panels. In the second case, the reverse link should be evaluated by replacing the time-packing scheme with a frequency-packing one. It is well known that NB-IoT uses SC-FDMA in the uplink and, as consequence, solutions that increase the spectral efficiency by overlapping the subcarriers should be better considered. As a side effect, SC-FDMA can compensate in part the increase of PAPR that overlapping in the time and frequency domains introduce. Finally, in the third case, the ISI that the process of shrinking the pulses/subcarriers introduces could be used as an artificial noise signal, which can mask the desired information from potential eavesdroppers.

## Figures and Tables

**Figure 1 sensors-21-03952-f001:**
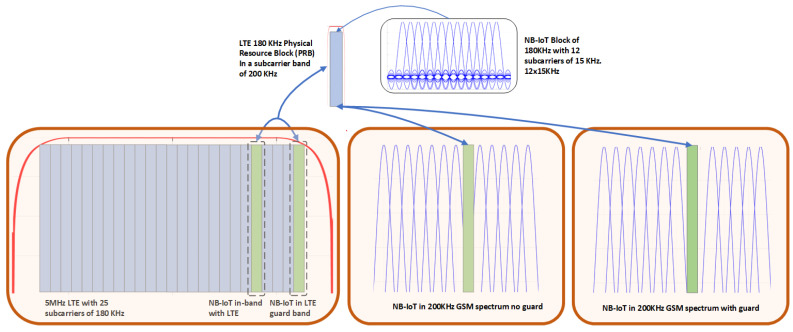
Typologies of NB-IoT deployment, namely stand-alone, in-band, and guard-band deployments. All these deployment configurations are compatible with the channelization that is used in contemporary mobile communication standards, such as GSM/2G (200 kHz channel) and LTE/4G (180 kHz Physical Resource Block).

**Figure 2 sensors-21-03952-f002:**
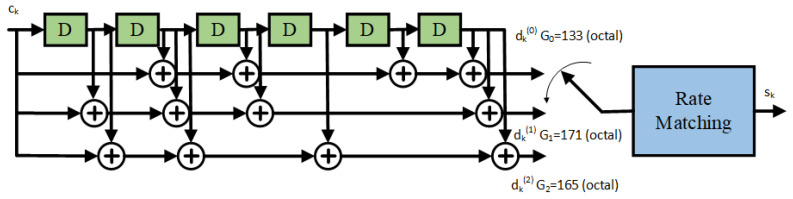
Overview of a rate 1/3 tail-biting convolutional coding with rate matching for NB-IoT [31].

**Figure 3 sensors-21-03952-f003:**
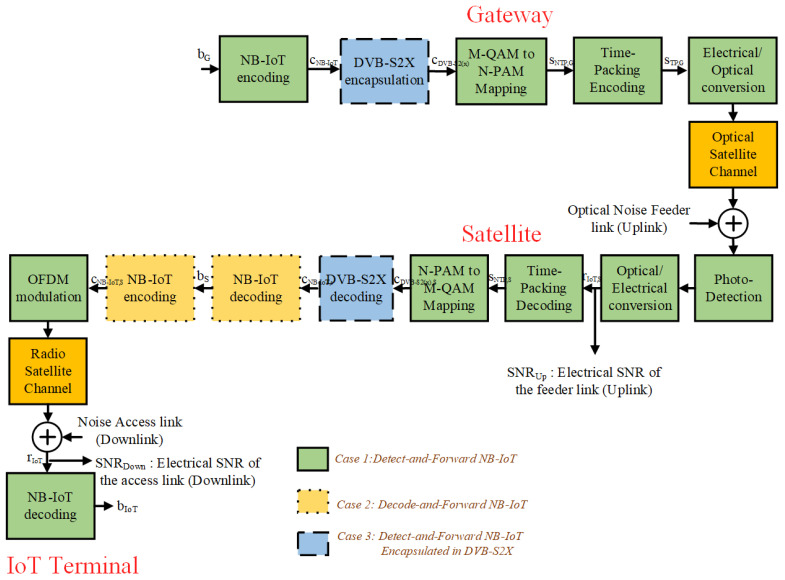
Block diagram of the GEO satellite forward link for the three relaying configurations under analysis: (1) Detect-and-Forward with NB-IoT (green blocks with solid edges); (2) Decode-and-Forward with NB-IoT (orange blocks with dotted edges); and (3) Detect-and Forward with NB-IoT/DVB-S2(X) (blue blocks with dashed edges).

**Figure 4 sensors-21-03952-f004:**
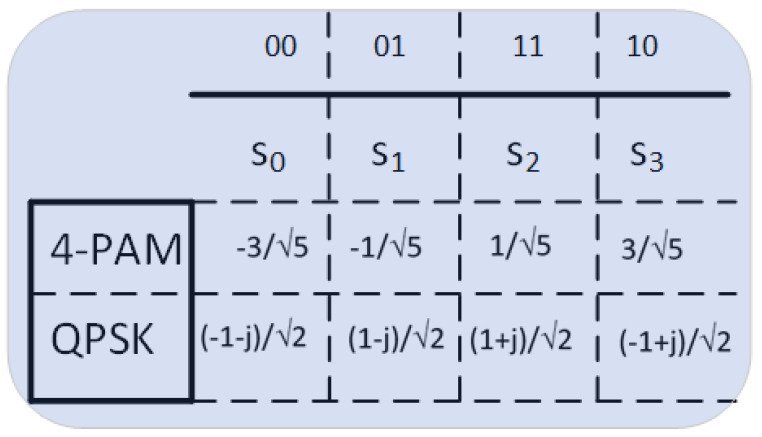
Gray mapping between 4-PAM and QPSK symbols performed in the satellite node.

**Figure 5 sensors-21-03952-f005:**
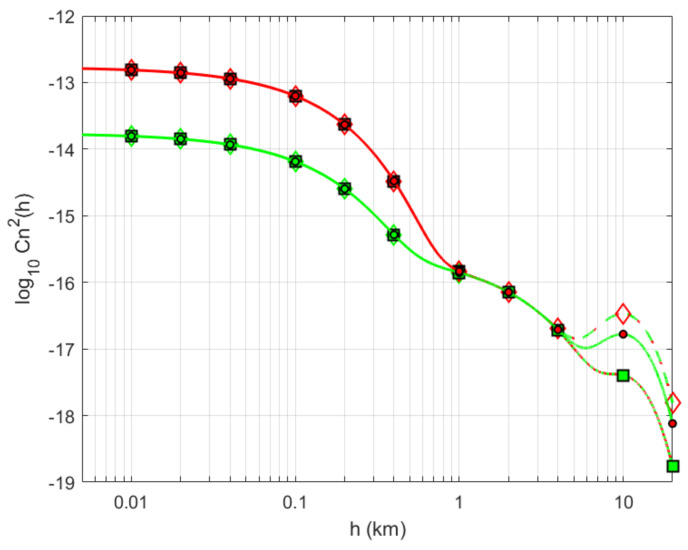
Refractive index structure $C_{n}^{2}\left(h\right)$ along the slant path for the H-V day model as a function of the altitude. Red lines: $A_{0}\;=\;1.7\times 10^{-13}$ m^−2/3^. Green lines: $A_{0}\;=\;1.7\times 10^{-14}$ m^−2/3^. Wind speed: v=10 m/s (dotted lines with squares); v=21 m/s (solid-lines with circles); and v=30 m/s (dashed lines with diamonds).

**Figure 6 sensors-21-03952-f006:**
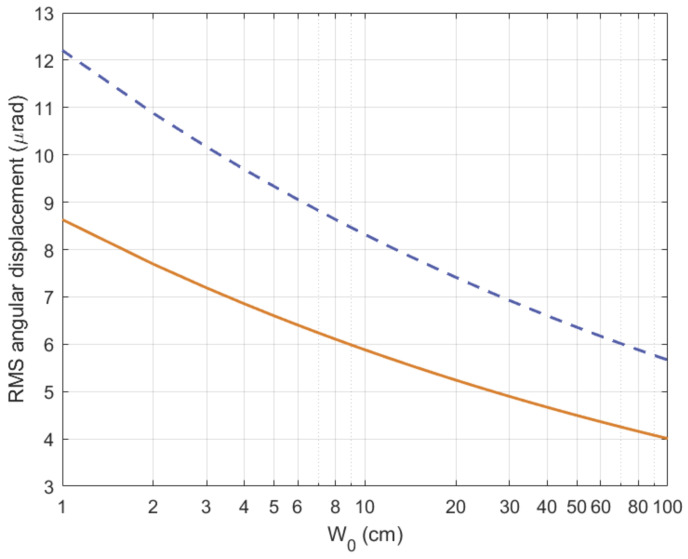
Root Mean Square (RMS) angular beam wander ($\sigma_{\mathrm{BW}}^{2}$) as a function of the beam radius ($W_{0}$) for a transmitter in the ground and a satellite in the space assuming λ=1.55μm and a refractive index structure following the H-V_5/7_ model (wind speed: v=21 m/s). Zenith angle: ζ=0 deg. (solid orange line) and ζ=60 deg. (dashed purple line).

**Figure 7 sensors-21-03952-f007:**
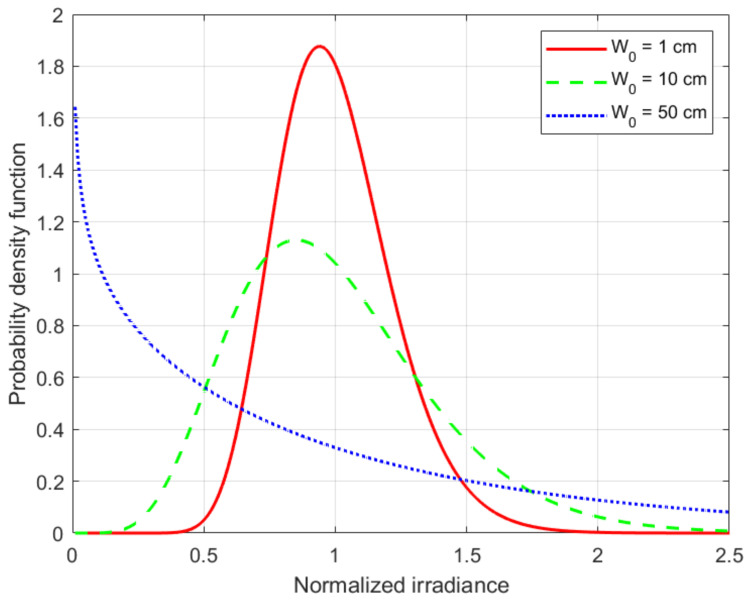
Gamma-Gamma probability density function for an untracked collimated beam plotted as a function of the normalized irradiance for a GEO optical feeder uplink channel with zenith angle ζ=0 deg. and H-V_5/7_ refractive index structure (λ=1.55μm, $r_{0}\;=\;19$ cm). Beam radius: $W_{0}\;=\;1$ cm (solid red lines), $W_{0}\;=\;10$ cm (dashed green lines), and $W_{0}\;=\;50$ cm (dotted blue lines).

**Figure 8 sensors-21-03952-f008:**
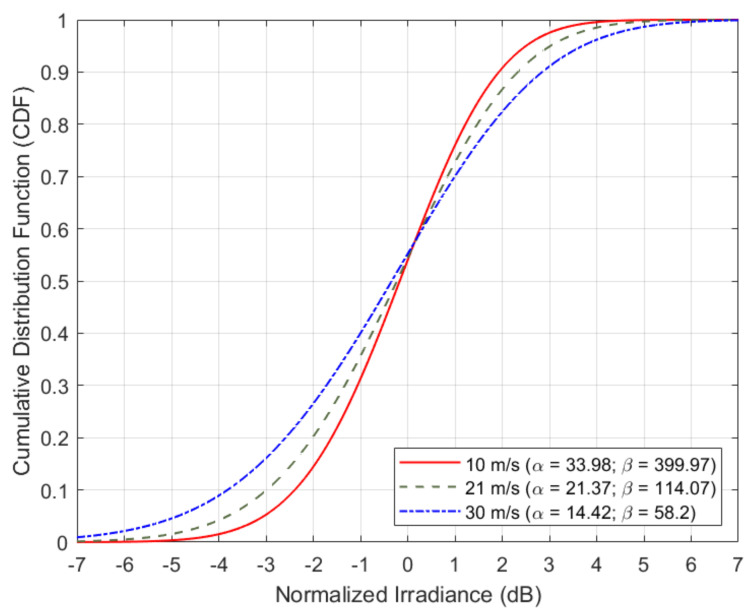
Cumulative distribution function of the power loss that turbulence-induced beam wander and scintillation introduces in the uplink direction of the optical feeder link in case of an untracked collimated beam with λ=1.55μm, $W_{0}=10$ cm, ζ=52.61 deg, $A_{0}=1.7\times 10^{-14}$ m^−2/3^. The ground station site is assumed next to Madrid ($h_{0}=864$ m) and the position of the GEO satellite is assumed at 19.2° East (*H* = 36,000 km). Wind speeds: v=10 m/s (solid red line), v=21 m/s (dashed green line), and v=30 m/s (dashed-dotted line).

**Figure 9 sensors-21-03952-f009:**
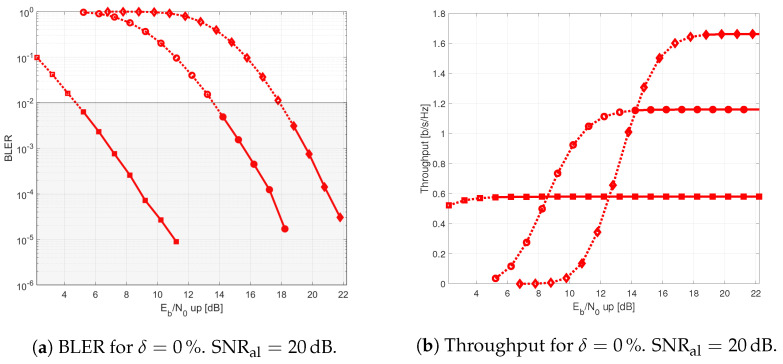
End-to-end BLER (left) and throughput (right) versus $E_{b}/N_{0}$ in the feeder link for a NB-IoT frame (${\mathrm{SNR}}_{\mathrm{al}}=20$ dB). NB-IoT code rates: 0.33333 (square), 0.66667 (circle) and 0.95556 (diamond), when the wind speed is v=21 m/s. Solid (dashed-) lines with (un)filled markers: $\mathrm{BLER}\leq 10^{-2}$ requirement (not) fulfilled. Diameter of telescope aperture: $W_{0}=21$ cm.

**Figure 10 sensors-21-03952-f010:**
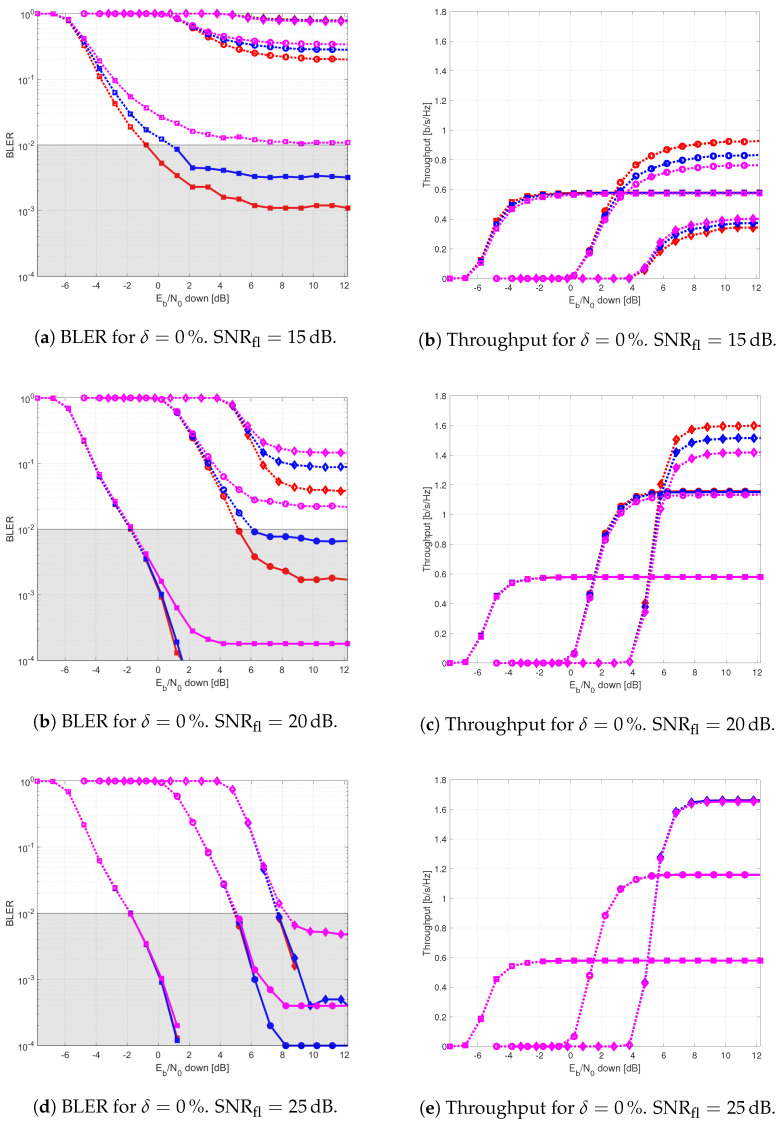
End-to-end BLER (left) and throughput (right) versus $E_{b}/N_{0}$ in the access link for a NB-IoT frame (${\mathrm{SNR}}_{\mathrm{fl}}=15$, 20, and 25 dB). NB-IoT code rates: 0.33333 (square), 0.66667 (circle), and 0.95556 (diamond), when the wind speed is v=10 m/s (red), v=21 m/s (blue) and v=30 m/s (magenta). Solid (dashed-) lines with (un)filled markers: $\mathrm{BLER}\leq 10^{-2}$ requirement (not) fulfilled. Diameter of telescope aperture: $W_{0}=10$ cm.

**Figure 11 sensors-21-03952-f011:**
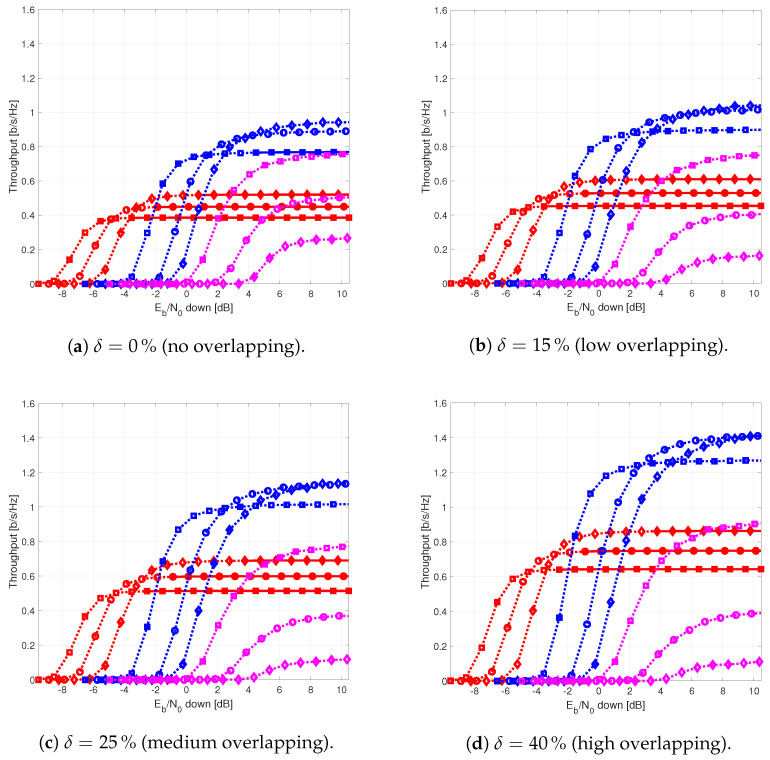
End-to-end throughput versus $E_{b}/N_{0}$ in the access link for a NB-IoT frame (${\mathrm{SNR}}_{\mathrm{fl}}=15$ dB). NB-IoT code rates: 0.2222 (red square), 0.259 (red circle), 0.3 (red diamond), 0.444 (blue square), 0.5319 (blue circle), 0.6 (blue diamond), 0.6386 (magenta square), 0.7678 (magenta circle), 0.8628 (magenta diamond). Solid (dashed-) lines with (un)filled markers: $\mathrm{BLER}\leq 10^{-2}$ requirement (not)fulfilled. Wind speed: v=21 m/s. Diameter of telescope aperture: $W_{0}=10$ cm.

**Figure 12 sensors-21-03952-f012:**
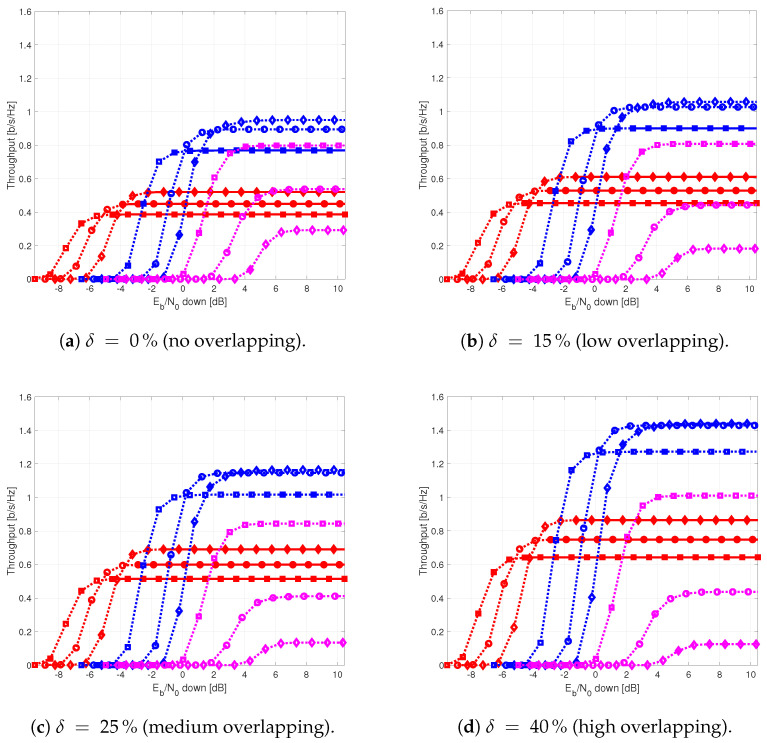
End-to-end throughput versus $E_{b}/N_{0}$ in the access link for a NB-IoT frame in a regenerative satellite (Decode-and-Forward with NB-IoT) (${\mathrm{SNR}}_{\mathrm{fl}}=15$ dB). NB-IoT code rates: 0.2222 (red square), 0.259 (red circle), 0.3 (red diamond), 0.444 (blue square), 0.5319 (blue circle), 0.6 (blue diamond), 0.6386 (magenta square), 0.7678 (magenta circle) and 0.8628 (magenta diamond). Solid (dashed-) lines with (un)filled markers: $\mathrm{BLER}\leq 10^{-2}$ requirement (not) fulfilled. Wind speed: v=21 m/s. Diameter of telescope aperture: $W_{0}=10$ cm.

**Figure 13 sensors-21-03952-f013:**
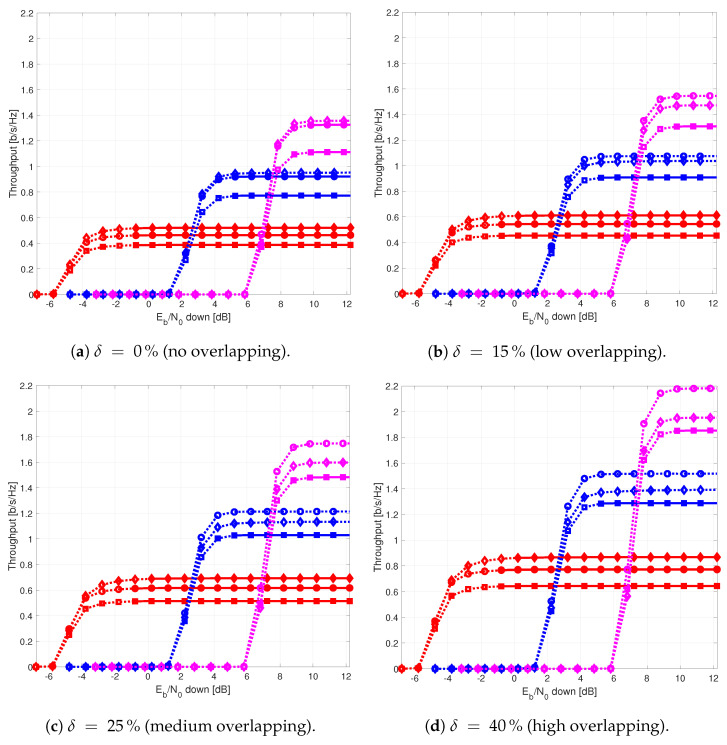
End-to-end throughput versus $E_{b}/N_{0}$ in the access link for a regenerative satellite that receives a NB-IoT frame encapsulated in a DVB-S2(X) satellite frame (${\mathrm{SNR}}_{\mathrm{fl}}=15$ dB). NB-IoT code rates: 1/3 (red), 2/3 (blue) and 0.96 (magenta). DVB-S2(X) code rates: 2/3 (squares), 0.8 (circles) and 0.9 (diamonds). Solid (dashed-) lines with (un)filled markers: $\mathrm{BLER}\leq 10^{-2}$ requirement (not) fulfilled. Wind speed: v=21 m/s. Diameter of telescope aperture: $W_{0}=10$ cm.

**Figure 14 sensors-21-03952-f014:**
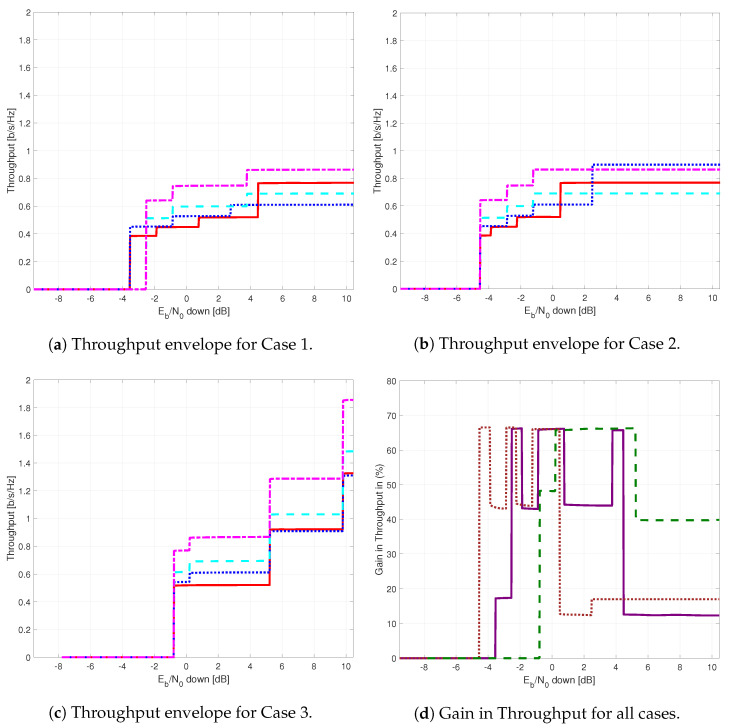
End-to-end throughput envelope (left and upper-right) and gain in the throughput envelope of time-packed versus no-time-packed (lower-right), represented as a function of the $E_{b}/N_{0}$ in the access link for Cases 1–3. Overlapping of δ=0% (solid red line), δ=15% (dotted blue line), δ=25% (dashed cyan line), and δ=40% (dashed-dotted magenta line). Figure 14d shows the gain in throughput of time-packed with respect to no-time-packed; here, Case 1 is represented with a continuous purple line, Case 2 is plotted with a dotted brown line, and Case 3 is shown with a dashed green line. Wind speed: v=21 m/s. Diameter of telescope aperture: $W_{0}=10$ cm.

**Figure 15 sensors-21-03952-f015:**
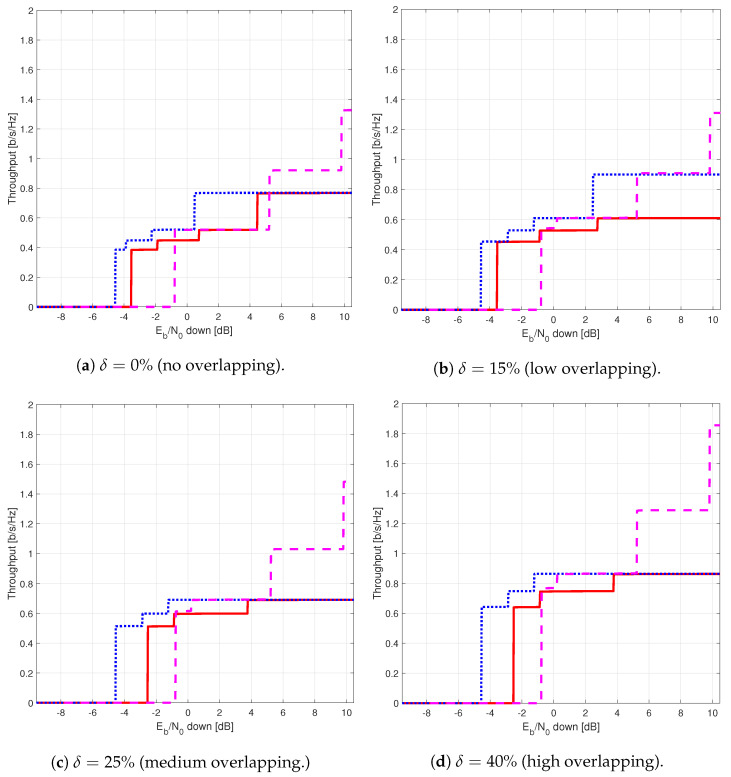
End-to-end envelope of throughput for all cases under study versus $E_{b}/N_{0}$ in the access link. Case 1 (continuous red line), Case 2 (dotted blue line) and Case 3 (dashed magenta line). Wind speed: v=21 m/s. Diameter of telescope aperture: $W_{0}=10$ cm.

**Figure 16 sensors-21-03952-f016:**
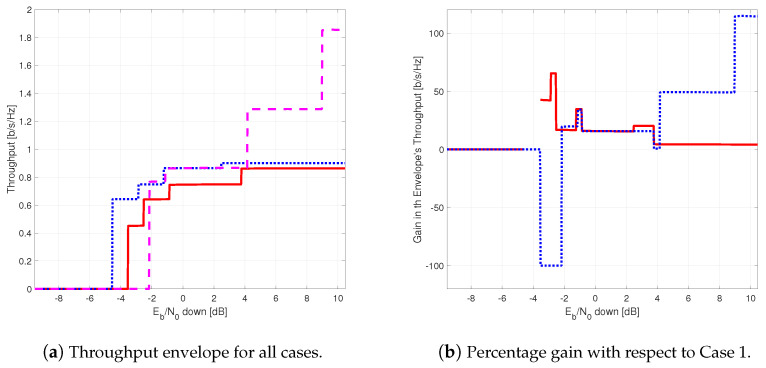
End-to-end throughput envelope for all cases under study versus $E_{b}/N_{0}$ in the access link. Case 1 (continuous red line), Case 2 (dotted blue line), and Case 3 (dashed magenta line) (left side). On the right, gain in the end-to-end throughput of Cases 2 (continuous red line) and 3 (dashed blue line) with respect to Case 1. Wind speed: v=21 m/s. Diameter of telescope aperture: $W_{0}=10$ cm.

**Figure 17 sensors-21-03952-f017:**
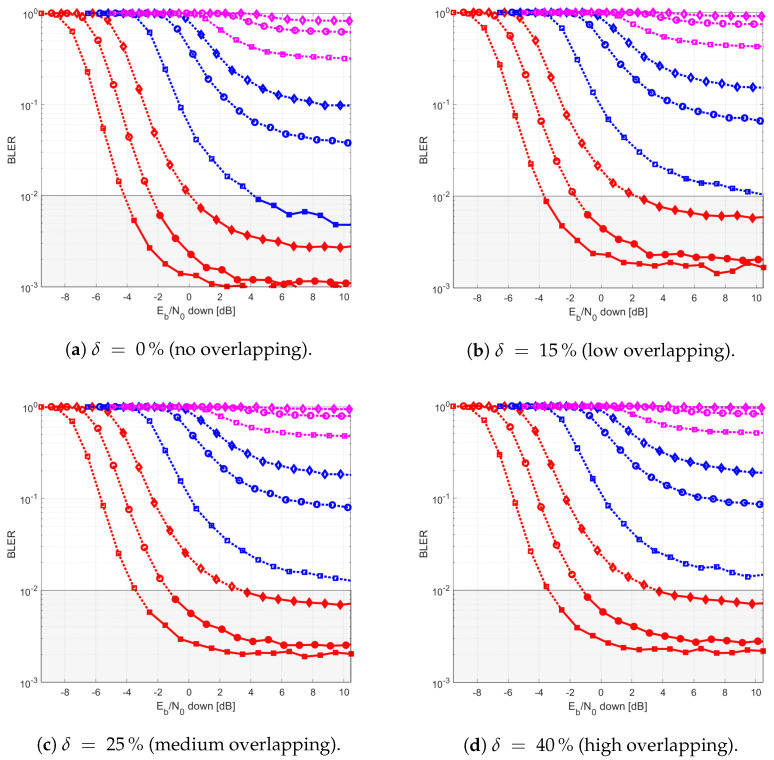
End-to-end BLER versus $E_{b}/N_{0}$ in the access link for a NB-IoT frame transmitted over a non-regenerative satellite (Detect-and-Forward) (${\mathrm{SNR}}_{\mathrm{fl}}=15$ dB). NB-IoT code rates: 0.2222 (red square), 0.259 (red circle), 0.3 (red diamond), 0.444 (blue square), 0.5319 (blue circle), 0.6 (blue diamond), 0.6386 (magenta square), 0.7678 (magenta circle) and 0.8628 (magenta diamond). Solid (dashed-) lines with (un)filled markers: $\mathrm{BLER}\leq 10^{-2}$ requirement (not) fulfilled. Wind speed: v=21 m/s. Diameter of telescope aperture: $W_{0}=10$ cm.

**Figure 18 sensors-21-03952-f018:**
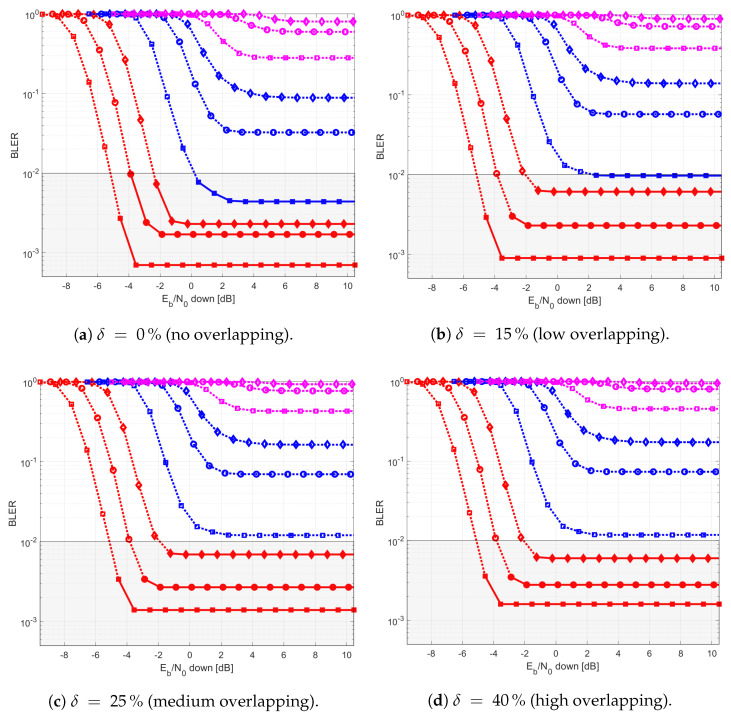
End-to-end BLER versus $E_{b}/N_{0}$ in the access link for a NB-IoT frame transmitted over a regenerative satellite (Decode-and-forward) (${\mathrm{SNR}}_{\mathrm{fl}}=15$ dB). NB-IoT code rates: 0.2222 (red square), 0.259 (red circle), 0.3 (red diamond), 0.444 (blue square), 0.5319 (blue circle), 0.6 (blue diamond), 0.6386 magenta square), 0.7678 (magenta circle) and 0.8628 (magenta diamond). Solid (dashed-) lines with (un)filled markers: $\mathrm{BLER}\leq 10^{-2}$ requirement (not) fulfilled. Wind speed: v=21 m/s. Diameter of telescope aperture: $W_{0}=10$ cm.

**Figure 19 sensors-21-03952-f019:**
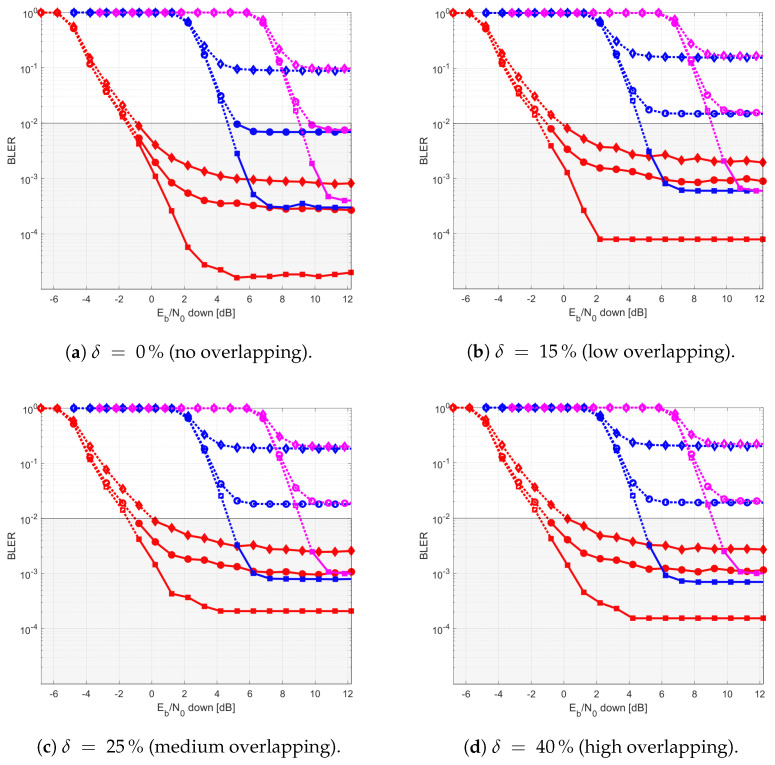
End-to-end BLER versus $E_{b}/N_{0}$ in the access link for a NB-IoT frame encapsulated in a DVB-S2(X) regenerative satellite (${\mathrm{SNR}}_{\mathrm{fl}}=15$ dB). NB-IoT code rates: 1/3 (red), 2/3 (blue) and 0.96 (magenta). DVB-S2(X) code rates: 2/3 (squares), 0.8 (circles) and 0.9 (diamonds). Solid (dashed-) lines with (un)filled markers: $\mathrm{BLER}\leq 10^{-2}$ requirement (not) fulfilled. Wind speed: v=21 m/s. Diameter of telescope aperture: $W_{0}=10$ cm.

**Table 1 sensors-21-03952-t001:** Closed-form expressions for the $LLR_{b_{0}}$ and $LLR_{b_{1}}$ of both 4-PAM and QPSK modulations.

${\mathrm{LLR}}_{b_{m}}$	4-PAM	QPSK
$LLR_{b_{0}}$	$4\;a\log\left(\frac{cosh(a-b)}{cosh(a+b)}\right)$	2Im(x)σn2
$LLR_{b_{1}}$	$-2\;b+\log\left(\frac{cosh(c)}{cosh(a)}\right)$	2Re(x)σn2

**Table 2 sensors-21-03952-t002:** Parameters of the optical feeder link used in the NB-IoT satellite system and associated power budget.

Symbol	Optical Link Parameter	Value	Unit
$P_{o,\mathrm{ld}}$	Optical power of LD (including EDFA booster)	47.0	dBm
$G_{o,\mathrm{tx}}$	Optical gain of transmitter (ground telescope)	110.9	dBi
$G_{o,\mathrm{rx}}$	Optical gain of receiver (satellite telescope)	112.8	dBi
$L_{o,\mathrm{fsl}}$	Free space loss of optical link (λ=1550 nm, *H* = 36,000 km)	289.8	dB
$L_{o,\mathrm{atm}}$	Atmospheric attenuation (absorption and scattering)	0–10	dB
$L_{o,\mathrm{bsl}}$	Beam spreading loss due to scintillation	1.6	dB
$L_{o,\mathrm{sys}}$	System losses in the optical feeder link	4.5	dB
$G_{\mathrm{edfa}}$	Gain of the optical amplifier (EDFA)	50.0	dB
μ	Responsivity of photodetector (PIN diode)	0.5	A/W
$B_{e}$	Bandwidth of electrical filter (PD output)	1.5	GHz
$B_{o}$	Bandwidth of optical channel (λ=1550 nm)	12.5	GHz
$\rho_{\mathrm{ase}}$	PSD of amplified spontaneous emissions	$2.0\times 10^{-19}$	W/Hz
$\rho_{\mathrm{rin}}$	PSD of RIN process (normalized)	−160	dBc/Hz
$\rho_{\mathrm{back}}$	PSD of background noise at EDFA input	$7.6\times 10^{-25}$	W/Hz
$i_{n}$	Electrical noise current spectral density	$1.0\times 10^{-11}$	A
$i_{\mathrm{dark}}$	Dark current at the PIN diode output	$1.0\times 10^{-10}$	A

**Table 3 sensors-21-03952-t003:** NB-IoT and DVB-S2(X) equivalent code rates for the relaying architectures under analysis.

$R_{c,\;\mathrm{dvb}}$	$R_{c,\;\mathrm{iot}}$	$R_{c,\;\mathrm{case}3}$	$R_{c,\;\mathrm{case}1/2}$
0.66667	0.33333	0.2222	0.2222
0.8	0.33333	0.2667	0.259
0.9	0.33333	0.3	0.3
0.66667	0.66667	0.4444	0.4444
0.8	0.66667	0.5333	0.5319
0.9	0.66667	0.6	0.6
0.66667	0.96	0.64	0.6386
0.8	0.96	0.768	0.7678
0.9	0.96	0.864	0.8628

**Table 4 sensors-21-03952-t004:** Maximum achievable throughput in terms of the overlapping factor and code rate.

$R_{c}$	δ = 0%	δ = 15%	δ = 25%	δ = 40%
0.2222	0.3864	0.4546	0.5152	0.6441
0.2590	0.4504	0.5299	0.6006	0.7507
0.3000	0.5217	0.6138	0.6957	0.8696
0.4444	0.7729	0.9093	1.0305	1.2881
0.5319	0.9250	1.0883	1.2334	1.5417
0.6000	1.0435	1.2276	1.3913	1.7391
0.6386	1.1106	1.3066	1.4808	1.8510
0.7678	1.3353	1.5709	1.7804	2.2255
0.8628	1.5005	1.7653	2.0007	2.5009

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
