# Peer review of "On the Use of NB-IoT over GEO Satellite Systems with Time-Packed Optical Feeder Links for Over-the-Air Firmware/Software Updates of Machine-Type Terminals"

_sensors, 2021, doi:10.3390/s21123952_

Round 1

Reviewer 1 Report

The paper reads well. A few suggestions:

1. The introduction is very confusing. It should be simple. The authors should answer the following: a) what is the problem that you are trying to solve? b) What are the state-of-the-art solutions for that? c) What are the shortcomings of these solutions? d) What approach you are taking to overcoming these shortcomings and how is that approach novel?

2. All the conditions stated in 3.1, this reviewer is not sure, how these are considered in evaluation section.

3. The conclusion section is confusing, it rather contains discussions. It will be appreciated if the authors include a separate discusisson section for results.

Author Response

Many thanks for your review. Your comments have been quite helpful to increase the quality of the paper.

Reviewer 2 Report

In my point of view several improvement are needed before this article is finalized for the publication.

  1. In order to attract the readers, motivation is highly important. Although authors have elaborated their motivation in introduction and related work sections. However it is suggested that it should be improved and highlighted the in introduction part.

2. The framework modeling and results should be evaluated more rigorously by making a comparative analysis along with discussion explaining why authors do think that others methods cannot comparatively perform well with the selected one.

3. Moreover proposed method, have been evaluated on a single dataset based on indoor temperature and light data, it is highly recommended to utilize another dataset as well to fully realize the performance of the proposed methods on different type of data as well.

4. I would highly recommend authors to explain the limitation of their work in a different section as well. Specially highlighting the limitations in NB-IoT over GEO satellite systems method.

5. English and spelling check requires 

Author Response

Many thanks for your review. They have been quite useful for us to improve the quality of this paper.

Reviewer 3 Report

The paper is interesting and technically novel, and it is overall written well.
There are some issues that should be addressed:
1. Is NB-IoT able to be used over MEO or LEO satellite systems? What is the difference between GEO, MEO, and LEO satellite systems?
2. Is the conclusion supported by real data?

Author Response

Many thanks for your comments. They have been quite useful for us to improve the quality of this paper.
